# ATMOSARENA: BENCHMARKING FOUNDATION MODELS FOR ATMOSPHERIC SCIENCES

## ABSTRACT

Deep learning has emerged as a powerful tool for atmospheric sciences, showing significant utility across various tasks in weather and climate modeling. In line with recent progress in language and vision foundation models, there are growing efforts to scale and finetune such models for multi-task spatiotemporal reasoning. Despite promising results, existing works often evaluate their model on a small set of non-uniform tasks, which makes it hard to quantify broad generalization across diverse tasks and domains. To address this challenge, we introduce `AtmosArena`, the first multi-task benchmark dedicated to foundation models in atmospheric sciences. `AtmosArena` comprises a suite of tasks that cover a broad spectrum of applications in atmospheric physics and atmospheric chemistry. To showcase the capabilities and key features of our benchmark, we conducted extensive experiments to evaluate two state-of-the-art deep learning models, ClimaX and Stormer on `AtmosArena`, and compare their performance with other deep learning and traditional baselines. By providing a standardized, open-source benchmark, we aim to facilitate further advancements in the field, much like open-source benchmarks have driven the development of foundation models for language and vision.

## 1 INTRODUCTION

Modeling of large-scale atmospheric systems is an omnipresent challenge for science and society. Traditionally, numerical methods are the dominating approach in atmospheric sciences, which operationalize rigorous systems of differential equations to simulate such phenomena (Lynch, 2008; Bauer et al., 2015). Despite their widespread use in practice, numerical methods suffer from many challenges, such as inadequate resolution of important small-scale physical processes and substantial computational demands (Balaji et al., 2017; Lavers et al., 2022; Leung et al., 2003; Rauscher et al., 2010). Deep learning has emerged as a powerful complement due to its ability to learn complex systems from historical data and produce fast predictions within seconds. Deep learning methods have proven great utility and performance across various atmospheric tasks, including but not limited to precipitation nowcasting (Ravuri et al., 2021b; Sønderby et al., 2020; Andrychowicz et al., 2023), medium-range weather forecasting (Weyn et al., 2020; Rasp & Thuerey, 2021; Keisler, 2022; Pathak et al., 2022b; Bi et al., 2022; Lam et al., 2023; Nguyen et al., 2023c; Chen et al., 2023b;a; Kochkov et al., 2023), climate projection (Watson-Parris et al., 2022b), climate downscaling (Baño Medina et al., 2020; Liu et al., 2020; Nagasato et al., 2021; Rodrigues et al., 2018; Sachindra et al., 2018; Vandal et al., 2019), air pollution forecasting (Ayturan et al., 2018; Bekkar et al., 2021; Tao et al., 2019; Bui et al., 2018; Heydari et al., 2022), and greenhouse gas emission prediction (Hamrani et al., 2020; Bakay & Ağbulut, 2021; Altikat, 2021).

Recent years have witnessed a paradigm shift from training task-specific models to developing foundation models for atmospheric sciences (Nguyen et al., 2023a; Bodnar et al., 2024), similar to models such as GPT-x (Brown et al., 2020; Achiam et al., 2023) in natural language processing, or CLIP (Radford et al., 2021) in computer vision. These foundation models are trained on large-scale and diverse datasets, enabling them to develop a rich, general understanding of the atmosphere. Once pre-trained, they can adapt efficiently to various downstream tasks, ranging from weather nowcasting to long-term climate projections, via lightweight finetuning. This approach is particularly attractive for atmospheric sciences, where there is an increasing availability of high-quality datasets and tasks have non-trivial global and regional structure.

Table 1: Comparisons between `AtmosArena` and existing works that consider multiple atmospheric tasks. `AtmosArena` offers the most comprehensive set of tasks, data, and evaluation metrics.

| Benchmark | Tasks | Data | Metrics |
|---|---|---|---|
| **AtmosArena** | Weather forecasting | ERA5 | RMSE, ACC |
| | S2S forecasting | ERA5 | RMSE, ACC, Spectral Div |
| | Climate data infilling | ERA5, Berkeley Earth | Bias, RMSE |
| | Climate model emulation | ClimateBench | Spatial, Global, Total, RMSE |
| | Climate downscaling | ERA5 | RMSE, Bias, Pearson |
| | Extreme weather events detection | ClimateNet | IoU, Precision, Recall, F-1 |
| **ClimateLearn** | Weather forecasting | ERA5 | RMSE, ACC |
| | Downscaling | ERA5 | RMSE, Bias, Pearson |
| | Projection | ClimateBench | Spatial, Global, Total, RMSE |
| **ClimaX** | Weather forecasting | ERA5 | RMSE, ACC |
| | S2S forecasting | ERA5 | RMSE, ACC |
| | Climate model emulation | ClimateBench | Spatial, Global, Total, RMSE |
| | Climate downscaling | ERA5 | RMSE, Bias, Pearson |
| **Aurora** | Weather forecasting | HRES Analysis | RMSE, ACC |
| | Air composition forecasting | CAMS Analysis | RMSE, ACC |

Standardized open-source benchmarks are crucial for the advancement of foundation models. In language, benchmarks such as HeLM (Liang et al., 2022), LLM Foundry, LM Evaluation Harness (Gao et al., 2023), and Big Bench (Srivastava et al., 2022) have aided researchers to systematically evaluate the performance of large language models. Similarly, for perception, comprehensive benchmarks such as VQA (Antol et al., 2015), SciBench (Wang et al., 2023), MMMU (Yue et al., 2023), and MathVista (Lu et al., 2023), have significantly accelerated research in multimodal foundation models. In stark contrast, there is no standardized multi-task benchmark for benchmarking atmospheric foundation models and existing works (Nguyen et al., 2023a; Bodnar et al., 2024) limit their evaluation to a relatively small set of non-overlapping tasks, which creates challenges in objective assessment of progress in the field.

To address this gap, we introduce `AtmosArena`, an open-source benchmark for foundation models in atmospheric sciences. To the best of our knowledge, `AtmosArena` is the first of its kind to offer a comprehensive evaluation framework tailored for this domain. `AtmosArena` encompasses a suite of tasks that span a wide spectrum of problems from both atmospheric and machine learning perspectives. Each task within `AtmosArena` is supported by datasets, fine-tuning protocols, evaluation code, standardized evaluation metrics, and a collection of deep learning and traditional baselines. This suite not only facilitates a fair assessment of model performance but also serves as a crucial tool for identifying opportunities for future development in the field. `AtmosArena` aims to set a new standard in the evaluation of atmospheric models, providing a solid foundation for the development of new methodologies. Table 1 summarizes the tasks, datasets, and metrics supported by `AtmosArena`.

To showcase the utility of `AtmosArena`, we conduct extensive experiments across all tasks included in the benchmark. We test and compare three representative classes of models: (1) deep learning with no pretraining, (2) single-source pretraining, and (3) multi-source pretraining. We also include traditional methods as simple baselines. To ensure fairness, we maintained consistent fine-tuning and evaluation settings across all models. The experimental results indicate that pretrained models generally outperform baselines without pretraining in most tasks. However, no single model consistently dominates across all tasks. This underscores the comprehensiveness of `AtmosArena` and highlights potential opportunities for future model development. In line with our commitment to openness and reproducibility, we will make all our data, code, and model checkpoints publicly available.

## 2 RELATED WORK

**Deep Learning for Atmospheric Sciences** Deep learning has revolutionized atmospheric sciences in recent years in both speed and accuracy. In weather forecasting, notable models like Pangu (Bi et al., 2022), Graphcast (Lam et al., 2023), and Stormer (Nguyen et al., 2023c) have surpassed the accuracy of the gold-standard IFS HRES system. This progress spans from simple models like ResNet (Rasp & Thuerey, 2021) to advanced architectures such as Graph Neural Networks (Keisler,

2022; Lam et al., 2023), Fourier neural operators (Pathak et al., 2022a), and Transformers (Bi et al., 2022; Nguyen et al., 2023a; Chen et al., 2023c;a; Nguyen et al., 2023c). In addition to medium-range, other works focus on forecasting at different time scales, such as nowcasting (Sønderby et al., 2020; Ravuri et al., 2021a; Andrychowicz et al., 2023) or longer-term prediction tasks (Watt-Meyer et al., 2023; Mouatadid et al., 2023). To account for uncertainty, recent works have also proposed ensemble forecasting with hybrid-physics models (Kochkov et al., 2024) or diffusion (Price et al., 2024), which are particularly useful for extreme event prediction like heavy rainfall (Zhang et al., 2023) and floods (Nearing et al., 2024).

**Foundation Models for Atmospheric Sciences** ClimaX (Nguyen et al., 2023a) is the first foundation model for weather and climate, pretrained on five simulated datasets from CMIP6 and finetuned on four downstream tasks. Aurora (Bodnar et al., 2024) is the latest atmospheric foundation model which scaled up pretraining to larger models, more data, and finer grid resolutions. Aurora was shown to achieve state-of-the-art performance in operational weather forecasting and air composition forecasting. In addition to atmospheric sciences, the development of scientific foundation models for physical domains is growing quickly as a field. For example, recent works in Partial Differential Equations (PDEs) modeling have proposed to pretrain large-scale models for micro-scale dynamical systems that can transfer in a zero-shot or few-shot fashion to unseen equations (Sun et al., 2024; Herde et al., 2024; Alkin et al., 2024; McCabe et al., 2023).

**Atmospheric Datasets and Benchmarks** Standardized benchmarks fuel the growth of atmospheric deep learning. WeatherBench (Rasp et al., 2020a; 2023) provides data, metrics, baselines, and a leaderboard for medium-range weather forecasting. Another common data source for weather forecasting is CMIP6 (Eyring et al., 2016b) which provides a large collection of simulation runs from climate models. SubseasonalClimateUSA (Mouatadid et al., 2024) and ChaosBench (Nathaniel et al., 2024) are two recent benchmarks that have been proposed to push the forecasting capabilities to sub-seasonal and seasonal time scales. Beyond forecasting, standard datasets have been developed for a diverse set of tasks in weather and climate, including climate emulation (Kaltenborn et al., 2023), sub-resolution physics modeling (Yu et al., 2024), precipitation prediction (de Witt et al., 2020; Sit et al., 2021), extreme weather events detection and localization (Rahnemoonfar et al., 2021; Requena-Mesa et al., 2021; Minixhofer et al., 2021; Prabhat et al., 2021; Racah et al., 2017), natural disaster-related tasks (Proma et al., 2022), atmospheric radiative transfer (Cachay et al., 2021), long-term global trends prediction (Watson-Parris et al., 2022b), cloud classification (Rasp et al., 2020b), nowcasting (Franch et al., 2020), tropical cyclone intensity prediction (Maskey et al., 2020), air quality metrics prediction (Betancourt et al., 2021), hydrometeorological time series analysis (Villaescusa-Navarro et al., 2022), and river flow analysis (Godfried et al., 2020). Beyond plain datasets, libraries such as ClimateLearn (Nguyen et al., 2023b), Scikit-downscale Hamman & Kent (2020), CCdownscaling Polasky et al. (2023), and CMIP6-Downscaling CarbonPlan (2022) provide software for training deep learning methods for various tasks in atmospheric sciences.

## 3 KEY COMPONENTS OF ATMOSARENA

As a first benchmark, we aim to build a comprehensive suite of tasks in atmospheric sciences, emphasizing diversity from both domain-specific and machine learning perspectives. Domain-wise, tasks are broadly classified into atmospheric physics or atmospheric chemistry. Atmospheric physics focuses on physical variables like temperature, humidity, and wind, essential for modeling weather patterns in the short-term and climate trends in the longer term. Atmospheric chemistry, on the other hand, focuses on the composition and transformation of atmospheric constituents, such as pollutants like carbon monoxide and dioxide, crucial for studying air quality and environmental health.

Due to space constraints, this section presents the six tasks under atmospheric physics: Medium-range Weather Forecasting, S2S Forecasting, Extreme Weather Events Detection, Climate Downscaling, Climate Data Infilling, and Climate Model Emulation. Tasks related to atmospheric chemistry are detailed in Appendix G. From a machine learning perspective, many common predictive tasks in atmospheric sciences can be mapped to well-defined problems in machine learning. Within this perspective, our benchmark can be seen as spanning five distinct categories of tasks defined on a grid: forecasting, segmentation, super-resolution, inpainting, and counterfactual prediction. This diverse suite of tasks allows us to obtain a holistic evaluation of atmospheric foundation models.

### 3.1 TASKS

**Medium-range weather forecasting** is the task of predicting the global weather conditions at a future time step $t + T$ given the weather conditions at or before the current step $t$, where the lead time $T$ ranges from a few hours to two weeks. A deep learning model takes an input of shape $V \times H \times W$ and outputs a prediction of shape $V' \times H \times W$, in which $V$ and $V'$ are the numbers of input and output atmospheric variables, respectively, while $H \times W$ denotes the spatial resolution of the data.

**Sub-seasonal-to-seasonal (S2S) forecasting** is similar to medium-range forecasting but with a longer lead time range between 2 weeks and 2 months (Vitart & Robertson, 2018; Vitart et al., 2022). This task bridges the gap between weather forecasting and climate modeling and holds significant socioeconomic value in disaster mitigation, but has received much less attention than the other two well-established tasks. Since the weather becomes too chaotic for any model to perform accurate point prediction after two weeks, we instead task the models to forecast the average statistics of key variables over a two-week window.

**Extreme weather events detection** is the task of identifying weather patterns that may lead to extreme weather events, such as tropical cyclones and atmospheric rivers. Deep learning models are trained to perform pixel-level detection and segmentation of these events in climate data. Specifically, the input typically consists of key atmospheric variables, and the output is a segmented map where each pixel is classified as part of an extreme event or as background. This approach allows for precise quantification of the frequency, intensity, and spatial extent of extreme events under various climate scenarios, providing valuable insights for climate research and policy-making.

**Climate downscaling** is the task of improving the spatial resolution of climate model outputs, which typically operate on large grid cells due to their high computational demands. This refinement is crucial for accurately representing local phenomena and informing regional policy decisions. In this task, deep learning models transform an input grid of dimensions $V \times H \times W$ into a higher-resolution output $V' \times H' \times W'$, where $H' > H$ and $W' > W$.

**Climate data infilling** involves estimating missing or incomplete data in historical and current climate datasets. This task aims to provide a more comprehensive and continuous historical record of important atmospheric variables, such as near-surface air temperature, enabling robust climate analysis and modeling. In data infilling, deep learning models are trained to predict missing values by leveraging patterns found in available data. The typical input to these models includes incomplete datasets of dimensions $V \times H \times W$, and the output is a complete dataset of the same dimensions, where the previously missing values are estimated by the model.

**Climate model emulation** involves predicting the annual mean global distributions of crucial climate variables like surface temperature and precipitation indices, given different scenarios of anthropogenic forcing factors such as carbon dioxide ($CO_2$) and methane ($CH_4$). The input is a tensor of shape $T \times V \times H \times W$ which captures the forcing conditions over $T$ consecutive years, and the output shape is $V' \times H \times W$. Unlike temporal forecasting, this task assesses a model's ability to predict the response of the climate system to varying levels of external factors, providing a foundation for long-term climate strategy and policy decisions.

### 3.2 DATASETS

**ERA5** maintained by ECMWF (Hersbach et al., 2020) is a common dataset for training and evaluating data-driven methods in atmospheric sciences (Bi et al., 2022; Lam et al., 2023; Nguyen et al., 2023c). ERA5 is a reanalysis dataset that provides the best guess of different climate variables at any point in time by integrating observational data with an advanced forecasting model known as the Integrated Forecasting System (IFS) (Wedi et al., 2015). ERA5 offers hourly data from 1979 to the present and at a $0.25°$ ($721 \times 1440$) global grid, totaling nearly $400,000$ data points at 37 different pressure levels. Given its extensive scale, we regrid the original data to $1.40625°$ ($128 \times 256$) grid and consider data from 1979 to 2020 for training and evaluation. We use ERA5 for four tasks in `AtmosArena`, including medium-range weather forecasting, S2S forecasting, climate downscaling, and data infilling.

**Berkeley Earth** provides a variety of high-quality temperature data products that incorporate a large set of temperature observations (Rohde & Hausfather, 2020). In `AtmosArena`, we use the global monthly average temperature data at $1°$ ($180 \times 360$) grid as an independent test dataset for the infilling task. We regrid the data to the common resolution of $1.40625°$.

**ClimateBench** is a benchmark for testing data-driven methods for climate model emulation (Watson-Parris et al., 2022b). ClimateBench consists of simulation outputs of the Norwegian Earth System Model (NorESM2) (Seland et al., 2020) from CMIP6 (Eyring et al., 2016a) that are run under different forcing scenarios for the period $2015 - 2100$. The dataset includes four input forcing factors – carbon dioxide ($CO_2$), sulfur dioxide ($SO_2$), black carbon (BC), and methane ($CH_4$), and the annual mean global distributions of four target variables – surface temperature, diurnal temperature range, precipitation, and the 90th percentile of precipitation.

**ClimateNet** is an expert-labeled dataset of tropical cyclones (TCs) and atmospheric rivers (ARs), two important weather patterns that may lead to extreme weather events (Prabhat et al., 2020). ClimateNet consists of $459$ data points of simulation runs of the Community Atmospheric Model (CAM5.1) from $1996 - 2013$. Each data point has a spatial resolution of $768 \times 1152$ with a total of 16 atmospheric variables, and each pixel is labeled with one of three classes – TCs, ARs, and Background.

## 3.3 Models

We consider a state-of-the-art representative from three classes of models. Many other recent models would also benefit from this benchmark Bodnar et al. (2024); Price et al. (2024), but they are currently closed-source. Table 2 shows the inference FLOPs and the number of parameters of each baseline we consider in this paper. We maintain a public leaderboard at `https://atmosarena.github.io/leaderboard/` to allow open and fair evaluation of both open- and closed-source models.

Table 2: FLOPs and model size of different baselines considered in `AtmosArena`.

|  | ClimaX | Stormer | UNet |
|---|---|---|---|
| FLOPs | 986.098B | 7377.751B | 969.404B |
| Parameter count | 110.842M | 468.752M | 577.745M |

**Non-pretrained model** We aim to provide state-of-the-art methods tailored to each specific task in `AtmosArena`. For tasks without an established baseline, we use UNet (Ronneberger et al., 2015) as the deep learning baseline. We chose UNet due to its excellent performance in various dense prediction tasks in computer vision, which resemble most of the atmospheric tasks in `AtmosArena`. The Unet models we train in the experiments have the same size of 500M parameters, for which we have performed extensive hyperparameters tuning to obtain a strong non-pretrained baseline.

**Single-source pretrained model** We include Stormer (Nguyen et al., 2023c), a state-of-the-art open-source deep learning model for medium-range weather forecasting. Stomer is a transformers-based architecture (Vaswani et al., 2017) trained on 6-hourly ERA5 data at $1.40625°$ resolution from 1979 to 2018. We chose Stormer since it was trained on the same spatial resolution as our datasets, and its simple architecture allows seamless finetuning on new tasks. Stormer has 400M parameters.

**Multi-source pretrained model** We include ClimaX (Nguyen et al., 2023a), the first large-scale atmospheric foundation model trained on multiple data sources. ClimaX was pretrained to perform temporal forecasting on five simulated datasets at $1.40625°$ from CMIP6 (Eyring et al., 2016a) and was shown to transfer well to various atmospheric tasks via finetuning. Since ClimaX and Stormer share similar transformer architectures and training objectives, comparing them helps examine if and when multi-source pretraining is beneficial to the model. ClimaX has 100M parameters.

## 3.4 Finetuning protocols

ClimaX and Stormer share a similar architecture, which consists of an embedding layer, a transformer backbone, and a prediction head. The embedding layer transforms an input of shape $V \times H \times W$ to a sequence of shape $(H/p \times W/p) \times D$, where $(H/p \times W/p)$ is the sequence length, $p$ is the patch size, and $D$ is the hidden dimension. The transformer backbone processes this sequence and outputs a sequence of the same shape, and finally the prediction head outputs a prediction of shape $V' \times H' \times W'$. We refer to the original papers for a detailed description of these models.

We consider two finetuning settings, one where we freeze the core transformer backbone, and the other where we finetune the entire network. The frozen setting helps examine the direct transferability of the

pretrained backbone to new tasks without further training. In tasks where the input or target variables were unseen during pretraining, we replace the pretrained embedding layer and prediction head with newly initialized networks. For datasets having a different spatial resolution from pretraining data, we interpolate the pretrained positional embedding to match the new sequence length.

# 4 BENCHMARK EVALUATION

This section evaluates different models on six atmospheric physics tasks described in Section 3.1. Through the experiments, we aim to showcase the breadth of `AtmosArena` and provide practical recommendations for finetuning atmospheric foundation models on new tasks. We refer to Appendix H for the atmospheric chemistry experiments. We also present infilling results on the Berkeley Earth dataset and regional case studies on S2S forecasting in Appendix H.

## 4.1 MEDIUM-RANGE WEATHER FORECASTING

We compare ClimaX and Stormer with Graphcast (Lam et al., 2023) – a leading forecasting method, and Climatology – a simple baseline, on weather forecasting with lead times from 1 to 14 days. We consider six target variables: temperature at 2 meters (T2m), zonal (U10m) and meridional (V10m) wind at 10 meters, geopotential at 500hPa (Z500), temperature at 850hPa (T850), and specific humidity at 700hPa (Q700), which are commonly used to verify forecasting models in previous works. Since Stormer and Graphcast were trained specifically for forecasting, we roll-out the pretrained checkpoints to obtain forecasts at different lead times without further training. For ClimaX, we perform full finetuning for each specific lead time and target variable, following the protocol in the original paper. All deep learning methods are trained on ERA5 from 1979 to 2018 and tested on 2020. The same data split is used for other tasks unless noted otherwise.

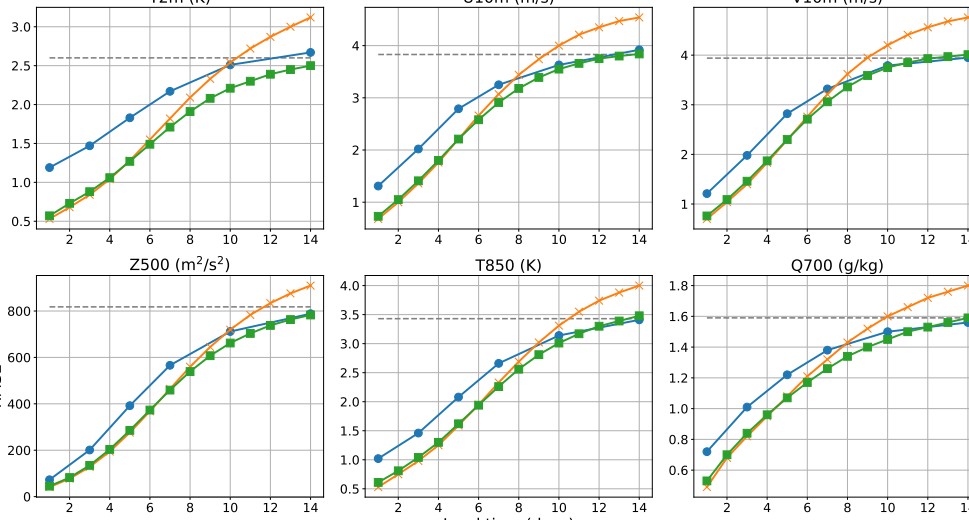

Figure 1: Medium-range weather forecasting performance measured by RMSE on six key variables at different lead times. Solid lines are deep learning models and the dashed line denotes the climatology baseline. Lower RMSE indicates better performance.

Figure 1 summarizes the RMSE results of this task (see Appendix for other metrics). Stormer is the best overall method, performing competitively with Graphcast at short lead times and much better at longer time scales. Graphcast works well for short lead times, but its performance degrades quickly and becomes worse than Climatology after day 10. ClimaX, on the other hand, performs poorly at small lead times but surpasses Graphcast at around day 10 and catches Stormer at day 14. This is because ClimaX performs direct forecasting which avoids error accumulation at long lead times.

## 4.2 SUBSEASONAL-TO-SEASONAL (S2S) FORECASTING

We evaluate ClimaX, Stormer, and Unet on forecasting the biweekly average statistics of four target variables – Z500, T850, T2m, and Q700. We consider two lead times of 2 weeks and 4 weeks, in which the average statistics are computed over weeks 3-4 and weeks 5-6, respectively. We construct the biweekly average data for training and evaluation from ERA5. For each baseline, we train two separate models to predict directly the average values at two different lead times. For ClimaX and Stormer, we consider two finetuning protocols where we either freeze (ClimaX frozen and Stormer frozen) or finetune (ClimaX finetuned and Stormer finetuned) the transformer backbone. Similar to medium-range weather forecasting, we include Climatology to examine if deep learning models achieve meaningful skills for S2S forecasting compared to this simple baseline.

Table 3: S2S performance measured by RMSE and ACC on four target variables at two lead times.

|  |  | Z500 | | T850 | | T2m | | Q700 | |
| --- | --- | --- | --- | --- | --- | --- | --- | --- | --- |
|  |  | Weeks 3-4 | Weeks 5-6 | Weeks 3-4 | Weeks 5-6 | Weeks 3-4 | Weeks 5-6 | Weeks 3-4 | Weeks 5-6 |
| **RMSE** ($\downarrow$) | ClimaX frozen | 458.53 | 471.58 | 1.79 | 1.84 | 1.67 | 1.73 | **0.69** | **0.70** |
|  | ClimaX finetuned | **453.05** | 469.92 | **1.77** | **1.80** | 1.65 | 1.70 | **0.69** | 0.71 |
|  | Stormer frozen | 461.19 | **467.37** | **1.77** | 1.81 | **1.56** | 1.69 | 0.70 | 0.72 |
|  | Stormer finetuned | 466.82 | 475.06 | 1.79 | 1.84 | 1.64 | 1.75 | 0.71 | 0.72 |
|  | Unet | 498.46 | 521.32 | 1.90 | 2.09 | 1.63 | 2.29 | 0.74 | 0.75 |
|  | Climatology | 475.58 | 475.58 | 2.00 | 2.00 | 1.61 | **1.61** | 0.76 | 0.76 |
| **ACC** ($\uparrow$) | ClimaX frozen | **0.84** | 0.81 | **0.92** | 0.90 | 0.96 | **0.95** | **0.86** | 0.84 |
|  | ClimaX finetuned | **0.84** | 0.81 | **0.92** | 0.90 | 0.95 | 0.94 | **0.86** | 0.84 |
|  | Stormer frozen | 0.78 | 0.77 | 0.88 | 0.87 | 0.95 | 0.94 | 0.81 | 0.81 |
|  | Stormer finetuned | 0.77 | 0.77 | 0.87 | 0.87 | 0.94 | 0.93 | 0.82 | 0.82 |
|  | Unet | **0.84** | **0.84** | **0.92** | **0.91** | **0.97** | 0.93 | 0.85 | **0.85** |

Table 3 summarizes the results of S2S forecasting. In terms of RMSE, both ClimaX and Stormer have meaningful skills except for T2m, while Unet underperforms Climatology for most variables. Interestingly, the frozen version of ClimaX and Stormer performs competitively to their fully finetuned counterpart. This result highlights the importance of pretraining, which allows models to efficiently transfer to new forecasting tasks without further training of the transformer backbone. In terms of ACC, ClimaX and Unet perform similarly while Stormer lags behind. Overall, ClimaX outperforms Stormer in this task despite having a poorer performance on medium-range weather forecasting. This can be explained by the difference between the pretraining objective of the two models, where ClimaX was trained to perform forecasting at much longer horizons (6 hours to 1 week) compared to Stormer (6 hours to 1 day).

## 4.3 CLIMATE DOWNSCALING

We consider the task of downscaling for six key variables: Z500, T850, T2m, Q700, U10m, and V10m. We use ERA5 at $5.625°$ as the low-resolution input, and ERA5 at $1.40625°$ as the high-resolution target, corresponding to $4\times$ upsampling. We include Unet as a deep learning baseline in addition to the two finetuning versions of ClimaX and Stormer. We report RMSE and Absolute Mean Bias, which is the absolute difference between the spatial mean of predictions and ground-truths.

Table 4: Downscaling performance measured by RMSE and Absolute Mean Bias on six variables.

|  |  | Z500 | T850 | T2m | Q700 | U10m | V10m |
| --- | --- | --- | --- | --- | --- | --- | --- |
| **RMSE** ($\downarrow$) | ClimaX frozen | 105.49 | 0.93 | 1.16 | 0.70 | 1.02 | 1.01 |
|  | ClimaX finetuned | 74.62 | 0.78 | 0.94 | 0.61 | 0.83 | 0.83 |
|  | Stormer frozen | 104.26 | 0.95 | 1.12 | 0.76 | 1.07 | 1.05 |
|  | Stormer finetuned | **38.84** | **0.57** | **0.62** | **0.55** | **0.64** | **0.64** |
|  | Unet | 47.65 | 0.66 | 0.73 | 0.56 | 0.70 | 0.70 |
| **Absolute Mean Bias** ($\downarrow$) | ClimaX frozen | 28.660 | 0.167 | 0.054 | **0.001** | 0.032 | 0.009 |
|  | ClimaX finetuned | 13.830 | 0.153 | 0.119 | 0.002 | **0.007** | **0.001** |
|  | Stormer frozen | 17.540 | **0.046** | 0.048 | **0.001** | 0.019 | 0.011 |
|  | Stormer finetuned | **0.090** | 0.051 | **0.031** | **0.001** | 0.011 | 0.017 |
|  | Unet | 8.790 | 0.140 | 0.040 | 0.005 | 0.011 | 0.006 |

Table 4 shows the performance of the considered methods. Unlike the forecasting tasks, there is a significant gap between the frozen and the fully finetuned models of ClimaX and Stormer. This

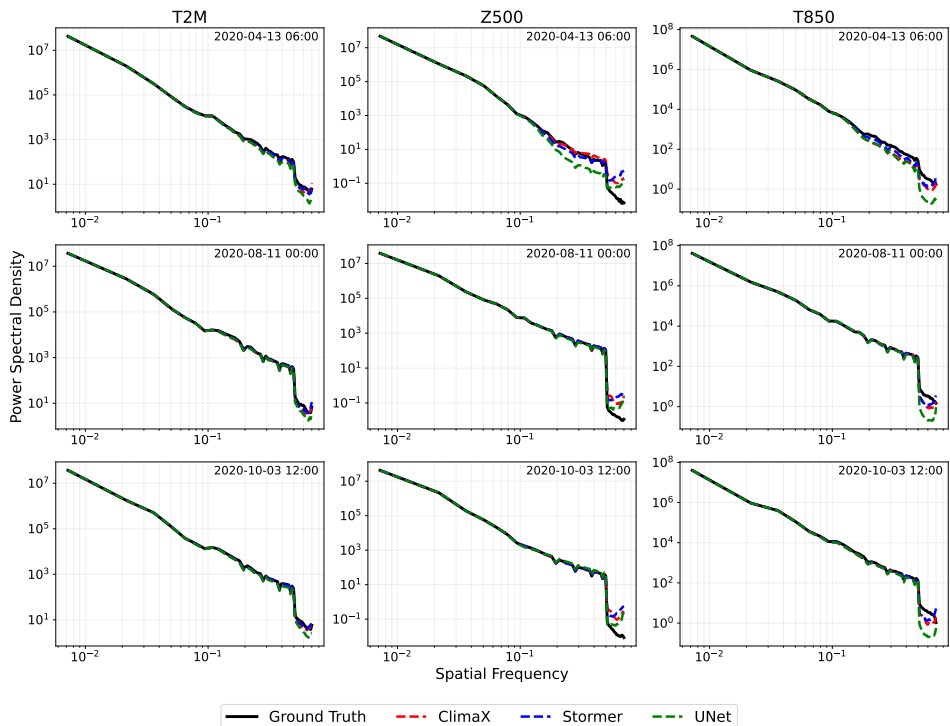

Figure 2: PSD plots of the baselines in comparison with the ground truths across three variables and three random test data points.

indicates that the transformer backbone pretrained for temporal forecasting might be sub-optimal for spatial downscaling and further finetuning is required to achieve good performance. Stormer is the best model in this task with the lowest RMSE and Absolute Mean Bias for most variables, followed by the Unet baseline. Since ClimaX has the lowest parameter count, we hypothesize that larger models tend to perform better in this task. This observation was also suggested by the scaling analysis in the original ClimaX paper.

In addition to quantitative metrics, we also plot the Power Spectral Density (PSD) to examine how well each model preserves the power spectrum across different spatial scales of the ground truth. To create these plots, we computed the 2D Power Spectral Density using the Fast Fourier Transform (FFT) for each spatial field, then performed radial averaging to obtain 1D PSD curves that show how power varies with spatial frequency. For each variable we consider (T2M, Z500, T850), we plotted the PSD curves of the ground truth and predictions from the three models on a log-log scale.

Figure 2 shows the PSD plots for three randomly selected data points in the test set. All three models display excellent agreement with the ground truth across low to medium spatial frequencies for all variables, indicating they accurately capture large-scale spatial patterns. However, there are notable differences at high spatial frequencies ($> 0.2$): UNet tends to underestimate the power at these frequencies, suggesting it may smooth out fine-scale details, while ClimaX and Stormer better preserve these high-frequency components. The results suggest that two pretrained models, ClimaX and Stormer, have an advantage in preserving fine-scale spatial details compared to UNet.

## 4.4 DATA INFILLING

We test the ability of foundation models to fill in missing temperature data, which is a common issue due to gaps in the coverage of observation stations. We construct training and validation data for this task from ERA5. During training, we generate a random mask for each training data point, with the mask ratio (missing ratio) drawn from a uniform distribution $r \sim \mathcal{U}[0.1, 0.9]$. We test each model to perform infilling with a set of mask ratios $r \in \{0.1, 0.3, 0.5, 0.7, 0.9\}$, where a fixed set of masks for each ratio is pre-generated and saved to disk to maintain evaluation consistency across models.

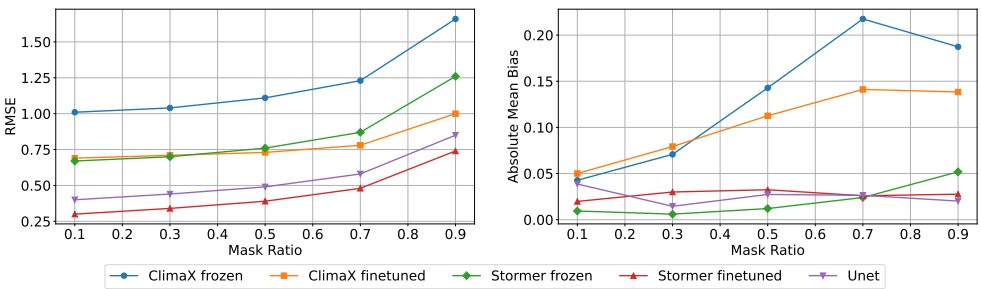

Figure 3: Infilling performance for surface temperature measured by RMSE and Absolute Mean Bias with different missing ratios.

Figure 3 shows the performance of the considered models for different mask ratios. Similar to downscaling, fully finetuned models work much better than frozen counterparts, and Stormer is the best method for this task. This result again highlights the difference between temporal and spatial tasks and the need for full finetuning to achieve good performance.

## 4.5 CLIMATE MODEL EMULATION

We aim to predict the annual mean global distributions of four target variables: surface air temperature, diurnal temperature range (difference between daily maximum and minimum surface air temperature), precipitation, and the 90th percentile precipitation. The input variables are four forcing factors: carbon dioxide ($CO_2$), sulfur dioxide ($SO_2$), black carbon (BC), and methane ($CH_4$). Following ClimateBench, we report $NRMSE_s$, $NRMSE_g$, and $NRMSE_t = NRMSE_s + 5\times NRMSE_g$ as the evaluation metrics. We use the best method in ClimateBench, namely ClimateBench-NN, as the baseline in addition to ClimaX and Stormer. We note that in this task, both the input and target variables were unseen during the pretraining of ClimaX and Stormer, so we replaced their embedding layer and prediction head with randomly initialized networks. Therefore, the transformer backbone essentially serves as a feature extractor. We finetune a separate model for each target variable.

Table 5: Climate model emulation performance measured by $NRMSE_s$, $NRMSE_g$, and $NRMSE_t$.

| | Surface air temperature | | | Diurnal temperature range | | | Precipitation | | | 90th percentile precipitation | | |
|---|---|---|---|---|---|---|---|---|---|---|---|---|
| | $NRMSE_s$ | $NRMSE_g$ | $NRMSE_t$ | $NRMSE_s$ | $NRMSE_g$ | $NRMSE_t$ | $NRMSE_s$ | $NRMSE_g$ | $NRMSE_t$ | $NRMSE_s$ | $NRMSE_g$ | $NRMSE_t$ |
| ClimaX frozen | **0.085** | **0.043** | **0.297** | **6.688** | **0.810** | **10.739** | 2.193 | 0.183 | 3.110 | **2.681** | 0.342 | **4.389** |
| ClimaX finetuned | 0.086 | **0.043** | 0.300 | 7.148 | 0.961 | 11.952 | 2.360 | 0.206 | 3.390 | 2.739 | 0.332 | 4.397 |
| Stormer frozen | 0.117 | **0.043** | 0.334 | 9.123 | 0.980 | 14.022 | 6.159 | 0.210 | 7.211 | 6.773 | 0.296 | 8.254 |
| Stormer finetuned | 0.126 | 0.047 | 0.361 | 8.598 | 0.834 | 12.767 | 6.180 | 0.391 | 8.136 | 6.797 | 0.316 | 8.376 |
| ClimateBench-NN | 0.123 | 0.080 | 0.524 | 7.465 | 1.233 | 13.632 | 2.349 | **0.151** | **3.104** | 3.108 | **0.282** | 4.517 |

Table 5 shows the superior performance of ClimaX in this task, outperforming Stormer and the ClimateBench-NN baseline by a large margin. This result highlights a unique benefit of multi-source pretraining in acquiring a general-purpose backbone that allows for easy transferability to downstream tasks and datasets significantly different from pretraining. Moreover, frozen models generally work better than the fully finetuned counterparts for this task. This can be explained by the small data size of ClimateBench (754 data points), so further finetuning of the backbone can lead to overfitting and hurt the test performance. A similar result was observed in the ClimaX paper.

## 4.6 EXTREME WEATHER DETECTION

Finally, we consider the task of detecting Tropical Cyclones (TCs) and Atmospheric Rivers (ARs), two atmospheric phenomena highly correlated with extreme weather events. We use the ClimateNet dataset for finetuning and evaluation, in which we use data from 1996 to 2010 for training and validation, and 2011 to 2013 for testing. We finetune ClimaX and Stormer to classify each pixel into one of three classes: TC, AR, and Background (BG). Similar to climate model emulation, we replace the pretrained embedding and prediction layer with randomly initialized networks. Since ClimateNet data is of much higher resolution, we increase the patch size to 8 for both ClimaX and Stormer, and interpolate the pretrained positional embedding to match the new sequence length.

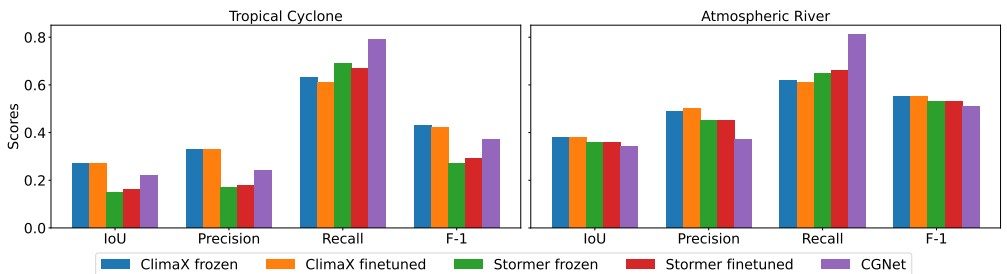

Figure 4: Extreme weather detection performance measured by IoU, Precision, Recall, and F-1.

Figure 4 compares the performance of ClimaX and Stormer with CGNet (Wu et al., 2020), a lightweight segmentation architecture based on CNN specifically designed for this task. Since the BG class dominates other classes, we adopt the weighted Jaccard loss function (Lacombe et al., 2023) to counter this class imbalance. The two finetuned versions of ClimaX work best in this task with respect to IoU and F-1, significantly outperforming its counterpart Stormer. This again demonstrates the importance of multi-source pretraining in obtaining higher transferable backbones. ClimaX also outperforms CGNet in $3/4$ metrics, showing the benefit of foundation models over specialized architectures.

## 5 CONCLUSION

We presented `AtmosArena`, the first benchmark dedicated to foundation models in atmospheric sciences. `AtmosArena` offers a diverse suite of tasks, datasets, and evaluation metrics to evaluate a foundation model holistically. `AtmosArena` not only provides a standard benchmark for comparing model performance but also serves as a crucial tool for identifying future research works. In addition, we release all our data, code, and model checkpoints, facilitating reproducible research and broadening collaborations. Given the vast development of scientific foundation models, we believe our contribution is timely and useful for both machine learning and atmospheric communities.

**Limitations and Future Work** With academic resource constraints, we acknowledge that there are various directions to improve `AtmosArena` in each of four dimensions – datasets, tasks, models, and evaluations. One such direction involves integrating regional datasets and expanding the collection of supported data sources. On the task side, we plan to include probabilistic tasks that are an important aspect of modeling weather and climate. For models and evaluations, we plan to find platforms for hosting atmospheric foundation models, along with an accompanying leaderboard.

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

# A APPENDIX

# B LICENSES AND TERMS OF USE

The source code will be available online under the MIT License upon acceptance. The licenses of the datasets we use in `AtmosArena` are as follows:

- ERA5 is curated and provided by WeatherBench2 which is licensed under Apache License 2.0 (`https://github.com/google-research/weatherbench2/blob/main/LICENSE`).

- Berkeley Earth (`https://berkeleyearth.org/data/`), ClimateBench (`https://zenodo.org/record/7064308`), ClimateNet (`https://gmd.copernicus.org/articles/14/107/2021/`) are available under the CC BY 4.0 license.

- CAMS Analysis provided by Copernicus Atmosphere Monitoring Service (CAMS) is free of charge, worldwide, non-exclusive, royalty-free and perpetual (`https://atmosphere.copernicus.eu/sites/default/files/repository/CAMS_data_license.pdf`).

- GEOS-CF (`https://portal.nccs.nasa.gov/datashare/gmao/geos-cf/`) provided by NASA is free for public access.

# C DATASETS

## C.1 DATASET DETAILS

Table 6: Summary of the datasets used to finetune and evaluate baselines in `AtmosArena`.

| Name | Resolution | Temporal coverage | Surface Variables | Multi-level Variables | Num levels | Size (GB) | Num frames |
|---|---|---|---|---|---|---|---|
| ERA5 | 128x256 | 1979-2020 | T2m, U10, V10, MSLP | Z, T, U, V, Q | 13 | 1600 | 61,324 |
| Berkeley Earth | 128x256 | 1850-2023 | T2m | N/A | N/A | 0.26 | 2,088 |
| ClimateBench | 32x64 | 2015-2100 | CO2, SO2, CH4, BC, TAS, DTR, PR, PR90 | N/A | N/A | 0.12 | 839 |
| ClimateNet | 768x1152 | 1996-2013 | TMQ, UBOT, VBOT, PS, PSL, PRECT, TS, TREFHT, ZBOT | U850, V850, QREFHT, T200, T500, Z1000, Z200 | N/A | 28 | 459 |
| CAMS Analysis | 128x256 | 2017-2022 | T2m, U10, V10, MSLP, TC CO, TC NO, TC NO2, TC SO2, TC O3, PM1, PM2.5, PM10 | U, V, T, Q, Z, CO, NO, NO2, SO2, O3 | 13 | 59 | 3774 |
| GEOS-CF | 128x256 | 2018-2023 | NO2, SO2, CO, O3, PM2.5 | N/A | N/A | 363 | 52,584 |

Table 6 details the datasets in `AtmosArena`, including their spatial resolution, temporal coverage, variables, and size. The full names of the abbreviated variables are:

- T2m, U10, V10, MSLP: 2-meter temperature, 10-meter zonal wind, 10-meter meridional wind, Mean sea level pressure.

- Z, T, U, V, Q: Geopotential, Temperature, Zonal wind, Meridional wind, Specific humidity at different pressure levels.

- CO2, SO2, CH4, BC: Carbon dioxide, Sulfur Dioxide, Methane, Black carbon.

- TAS, DTR, PR, PR90: Surface air temperature, Diurnal temperature range, Precipitation, 90th percentile precipitation.

- TMQ, UBOT, VBOT, PS, PSL, PRECT, TS, TREFHT, ZBOT: Total Precipitable Water, Lowest Model Level Zonal Wind, Lowest Model Level Meridional Wind, Surface Pressure, Sea Level Pressure, Total Precipitation Rate, Surface Temperature, Reference Height Temperature, Lowest Model Level Height.

- U850, V850, QREFHT, T200, T500, Z1000, Z200: Zonal Wind at 850 mb, Meridional Wind at 850 mb, Specific Humidity at Reference Height, Temperature at 200 mb, Temperature at 500 mb, Geopotential Height at 1000 mb, Geopotential Height at 200 MB.

- TC CO, TC NO, TC NO2, TC SO2, TC O3, PM1, PM2.5, PM10: Total Column Carbon Monoxide, Total Column Nitric Oxide, Total Column Nitrogen Dioxide, Total Column Sulfur Dioxide, Total Column Ozone, Particulate Matter 1um, Particulate Matter 2.5um, Particulate Matter 10um.
- CO, NO, NO2, SO2, O3: Zonal Wind, Meridional Wind, Temperature, Specific Humidity, Geopotential Height, Carbon Monoxide, Nitric Oxide, Nitrogen Dioxide, Sulfur Dioxide, Ozone.

For ERA5, following WeatherBench2 (Rasp et al., 2023), we used the 6-hourly subsampled data from the original ERA5 at 00:00, 06:00, 12:00, and 18:00, and used the 13 pressure levels for the multi-level variables: 50, 100, 150, 200, 250, 300, 400, 500, 600, 700, 850, 925, 1000. We use the same pressure levels for CAM Analysis. We also note that the resolutions of ERA5, Berkeley Earth, ClimateBench, CAMS Analysis, and GEOS-RF used in our paper are different from their original resolutions. We used bilinear interpolation to regrid the original data to the resolutions in Table 6.

## C.2 TRAIN, VALIDATION, AND TEST SPLIT

Table 7: Summary of train, validation, and test split of the datasets in `AtmosArena`.

| Name | Train time frame | Validation time frame | Test Year(s) |
|---|---|---|---|
| time frame | 1979-2018 | 2019 | 2020 |
| Berkeley Earth | N/A | N/A | 2000-2024 |
| ClimateBench | 2015-2100 | 2015-2100 | 2015-2100 |
| ClimateNet | 1996-2007 | 2008-2010 | 2011-2013 |
| CAMS Analysis | 2018-2020 | 2021 | 2022 |
| GEOS-CF | 2017-2020 | 2021 | 2022 |

Tabel 7 summarizes the train, validation, and test split of the datasets we included in `AtmosArena`. Most datasets are split according to time, where training, validation, and test data belong to non-overlapping time periods. For ClimateBench, which we used for the climate model emulation task, however, the data is split according to different future emission scenarios. We refer to ClimateBench (Watson-Parris et al., 2022b) for a detailed discussion of these scenarios.

## D EVALUATION METRICS

This section presents the formulation of the evaluation metrics we included in `AtmosArena`. We use the following notations across the metrics:

- $N$ is the number of data points
- $H$ is the number of latitude coordinates.
- $W$ is the number of longitude coordinates.
- $X$ and $\tilde{X}$ are the ground-truth and prediction, respectively.

Each equation below is computed for one single variable. To account for the non-uniformity of the grid cell areas when gridding a round Earth, most metrics are latitude-weighted to give more weight to the cells closer to the equator. The latitude weighting function is given by

$$L(i) = \frac{\cos(H_i)}{\frac{1}{H} \sum_{i=1}^{H} \cos(H_i)} \tag{1}$$

### D.1 FORECASTING METRICS

**Root Mean Square Error (RMSE)**

$$\text{RMSE} = \frac{1}{N} \sum_{k=1}^{N} \sqrt{\frac{1}{H \times W} \sum_{i=1}^{H} \sum_{j=1}^{W} L(i)(\tilde{X}_{k,i,j} - X_{k,i,j})^2}. \tag{2}$$

**Anomaly Correlation Coefficient (ACC)** is the spatial correlation between prediction anomalies $\tilde{X}^{'}$ relative to climatology and ground truth anomalies $X^{'}$ relative to climatology:

$$\text{ACC} = \frac{\sum_{k,i,j} L(i) \tilde{X}^{'}_{k,i,j} X^{'}_{k,i,j}}{\sqrt{\sum_{k,i,j} L(i) \tilde{X}^{'2}_{k,i,j} \sum_{k,i,j} L(i) X^{'2}_{k,i,j}}}, \tag{3}$$

$$\tilde{X}^{'} = \tilde{X} - C, X^{'} = X - C, \tag{4}$$

in which climatology $C$ is the temporal mean of the ground truth data over a fixed period. We used the climatology data from WeatherBench2 (Rasp et al., 2023) in our all experiments.

**Spectral Divergence (SpecDiv)** is inspired by KL divergence, which computes the expectation of the logarithmic ratio between the ground truth and predicted spectra. This metric emphasizes the relative error between the frequency components of the ground truth and prediction:

$$\text{SpecDiv} = \sum_k S'(k) \cdot \log\left(\frac{S'(k)}{\tilde{S}'(k)}\right) \tag{5}$$

where $S'(k)$ and $\hat{S}'(k)$ represent the spectral components of the ground truth and predictions, respectively, and $k$ denotes the spectral component.

## D.2 CLIMATE DOWNSCALING AND INFILLING METRICS

**Root Mean Square Error (RMSE)** This is the same as Equation (2).

**Mean Bias** measures the mean difference between the prediction and the ground truth. A positive mean bias shows overestimation, while a negative mean bias shows underestimation:

$$\text{Mean bias} = \frac{1}{N \times H \times W} \sum_{k=1}^{N} \sum_{i=1}^{H} \sum_{j=1}^{W} (\tilde{X}_{k,i,j} - X_{k,i,j}) \tag{6}$$

**Anomaly Pearson Coefficient** measures the Pearson correlation between the prediction and the ground truth anomalies. We first flatten the prediction and ground truth anomalies, and compute the metric as follows:

$$\rho_{\tilde{X}', X'} = \frac{\text{cov}(\tilde{X}', X')}{\sigma_{\tilde{X}'} \sigma_{X'}} \tag{7}$$

NOTE: For the Climate data infilling task, we compute the metrics over the masked cells only.

## D.3 CLIMATE MODEL EMULATION METRICS

**Normalized spatial root mean square error (NRMSE$_s$)** measures the spatial discrepancy between the temporal mean of the prediction and the temporal mean of the ground truth:

$$\text{NRMSE}_s = \sqrt{\left\langle \left(\frac{1}{N}\sum_{k=1}^{N}\tilde{X} - \frac{1}{N}\sum_{k=1}^{N}X\right)^2 \right\rangle} \bigg/ \frac{1}{N}\sum_{k=1}^{N}\langle X\rangle, \tag{8}$$

in which $\langle A \rangle$ is the global mean of $A$:

$$\langle A \rangle = \frac{1}{H \times W} \sum_{i=1}^{H} \sum_{j=1}^{W} L(i) A_{i,j} \tag{9}$$

**Normalized global root mean square error (NRMSE$_g$)** measures the discrepancy between the global mean of the prediction and the global mean of the ground truth:

$$\text{NRMSE}_g = \sqrt{\frac{1}{N}\sum_{k=1}^{N}\left(\langle\tilde{X}\rangle - \langle X\rangle\right)^2} \bigg/ \frac{1}{N}\sum_{k=1}^{N}\langle X\rangle. \tag{10}$$

**Total normalized root mean square error (Total)** is the weighted sum of NRMSE$_s$ and NRMSE$_g$:

$$\text{Total} = \text{NRMSE}_g + \alpha \cdot \text{NRMSE}_g, \tag{11}$$

where $\alpha$ is chosen to be 5 as suggested by Watson-Parris et al. (2022a).

## D.4 EXTREME WEATHER EVENTS DETECTION METRICS

Each pixel in the $H \times W$ grid is classified into one of three classes, leading to a confusion matrix per class (AR, TC, and BG). The performance metrics, calculated for each class, are defined as follows using the elements of the confusion matrix—True Positives (TP), False Positives (FP), True Negatives (TN), and False Negatives (FN):

**Intersection over Union (IoU)**

$$\text{IoU}_c = \frac{\text{TP}_c}{\text{TP}_c + \text{FP}_c + \text{FN}_c} \tag{12}$$

**Precision**

$$\text{Precision}_c = \frac{\text{TP}_c}{\text{TP}_c + \text{FP}_c} \tag{13}$$

**Recall**

$$\text{Recall}_c = \frac{\text{TP}_c}{\text{TP}_c + \text{FN}_c} \tag{14}$$

**F-1 Score**

$$\text{F-1}_c = 2 \times \frac{\text{Precision}_c \times \text{Recall}_c}{\text{Precision}_c + \text{Recall}_c} \tag{15}$$

**Specificity**

$$\text{Specificity}_c = \frac{\text{TN}_c}{\text{TN}_c + \text{FP}_c} \tag{16}$$

## E EXPERIMENT DETAILS

This section details the experiments we conducted in Section 3, including model architectures and hyperparameters, training objectives, and optimization.

### E.1 MODEL ARCHITECTURES

**Unet** We borrow our Unet implementation from PDEArena (Gupta & Brandstetter, 2022). Table 8 shows hyperparameters of Unet we use in all our experiments. The Unet model has a total of 500M parameters.

Table 8: Default hyperparameters of UNet

| Hyperparameter | Meaning | Value |
|---|---|---|
| Padding size | Padding size of each convolution layer | 1 |
| Kernel size | Kernel size of each convolution layer | 3 |
| Stride | Stride of each convolution layer | 1 |
| Channel multiplications | Determine the number of output channels for Down and Up blocks | $[1, 2, 2, 4]$ |
| Blocks | Number of Resnet blocks | 2 |
| Use attention | If use attention in Down and Up blocks | False |

**ClimaX and Stormer** For ClimaX and Stormer, we borrow the implementation from their original papers (Nguyen et al., 2023a;c), which we refer to for a detailed description of their architectures. Table 9 shows hyperparameters of ClimaX and Stormer we use in all our experiments. The parameter count for ClimaX and Stormer is 100M and 400M, respectively.

Table 9: Default hyperparameters of ClimaX and Stormer

| Hyperparameter | Meaning | ClimaX | Stormer |
|---|---|---|---|
| $p$ | Patch size | 4 | 2 |
| $D$ | Embedding dimension | 1024 | 1024 |
| Depth | Number of ViT blocks | 8 | 24 |
| # heads | Number of attention heads | 16 | 16 |
| MLP ratio | Determine the hidden dimension of the MLP layer in a ViT block | 4 | 4 |
| Prediction depth | Number of layers of the prediction head | 2 | 1 |
| Hidden dimension | Hidden dimension of the prediction head | 1024 | N/A |

### E.1.1 EXTENSIONS FOR CLIMATE MODEL EMULATION

We modify the architecture of ClimaX and Stormer for this task to account for the time dimension $T$ in the input. Each time slice of the input goes through the embedding layer and the transformer blocks independently, resulting in an output tensor of shape $T \times h \times w \times D$ where $D$ is the embedding dimension. This tensor then goes through a global pooling layer along the spatial dimensions $h$ and $w$, outputting a tensor of shape $T \times D$. This sequence of tensors is aggregated by a cross-attention layer over the time dimension to a single vector of $D$ dimensions. Finally, a linear layer predicts the output of shape $V \times H \times W$. The cross-attention layer along the time dimension is randomly initialized and trained together with the new embedding and prediction layer, as well as the transformer backbone.

### E.1.2 EXTENSIONS FOR EXTREME WEATHER EVENTS DETECTION

Since the spatial resolution of ClimateNet is $768 \times 1152$, training the original ClimaX and Stormer with patch sizes of 4 and 2, respectively, is too computationally expensive. To address this issue, we use a stack of 6 convolutional layers to embed the input before the attention blocks which outputs a tensor of shape $96 \times 144 \times D$, reducing the spatial resolution by 8. This tensor goes through the transformer blocks and a linear prediction head which outputs a tensor of shape $3 \times 96 \times 144$ where 3 is the number of classes. Finally, this output is bilinearly interpolated to the original spatial resolution of $768 \times 1152$. The bilinear interpolation module is also used by the baseline CGNet (Wu et al., 2020).

## E.2 TRAINING DETAILS

### E.2.1 DATA NORMALIZATION

For tasks that utilize ERA5 for training and evaluation, including medium-range weather forecasting, S2S forecasting, climate downscaling, and climate data infilling, we normalize both input and output variables to have mean 0 and standard deviation 1. The normalization constants are computed across the entire training set. During evaluation, predictions and ground-truths are de-normalized to the original scale before computing the metrics.

For the extreme weather events detection task that uses ClimateNet, we normalize the input variables similarly to ERA5, but not the output variables since they are discrete labels.

For the climate model emulation task that uses ClimateBench, we normalize the input variables similarly to ERA5, but not the output variables since we predict each target variable separately.

### E.2.2 TRAINING OBJECTIVES

**Regression** For the five regression tasks, including medium-range weather forecasting, S2S forecasting, climate downscaling, climate data infilling, and climate model emulation, we use the same latitude-weighted mean-squared error loss for training:

$$\mathcal{L}(\theta) = \frac{1}{V' \times H \times W} \sum_{v=1}^{V'} \sum_{i=1}^{H} \sum_{j=1}^{W} L(i)(\tilde{X}^{v,i,j} - X^{v,i,j})^2. \tag{17}$$

**Classification** For the extreme weather events detection task, we utilize the weighted Jaccard loss proposed in Lacombe et al. (2023) to prioritize the TC and AR classes:

$$\mathcal{L}(\theta) = \frac{1}{C \times H \times W} \sum_{c=1}^{C} \sum_{i=1}^{H} \sum_{j=1}^{W} \left( 1 - w_c \frac{\tilde{X}^{c,i,j} X^{c,i,j}}{(\tilde{X}^{c,i,j} + X^{c,i,j}) - \tilde{X}^{c,i,j} X^{c,i,j}} \right), \quad (18)$$

in which $w_c$ is the weight of class $c$. Following Lacombe et al. (2023), we set $w_c$ to 0.678, 31.08, and 2.9 for BG, TC, and AR, respectively.

### E.2.3  OPTIMIZATION

For all tasks, we used AdamW with parameters ($\beta_1 = 0.9, \beta_2 = 0.95$) and weight decay of $1e-5$ for all parameters except for the positional embedding in ClimaX and Stormer. We trained each model for 50 epochs with a batch size of 32, using a linear warmup schedule for 5 epochs, followed by a cosine-annealing schedule for 45 epochs. Table 10 shows the peak learning rate for each task.

Table 10: Learning rate for finetuning ClimaX in different downstream tasks

| Task | Finetuning LR | Scratch Training LR |
|------|---------------|---------------------|
| Medium-range weather forecasting | $5e-6$ | $5e-4$ |
| S2S forecasting | $5e-5$ | $5e-4$ |
| Climate downscaling | $5e-5$ | $5e-4$ |
| Climate data infilling | $1e-4$ | $5e-4$ |
| Climate model emulation | $5e-4$ | $5e-4$ |
| Extreme weather events detection | $5e-4$ | $5e-4$ |

For finetuning ClimaX and Stormer, we used a smaller learning rate for tasks that are similar to pretraining and a larger learning rate for tasks that are more different.

### E.2.4  SOFTWARE AND HARDWARE STACK

We use PyTorch (Paszke et al., 2019), `numpy` (Harris et al., 2020) and `xarray` (Hoyer & Hamman, 2017) to manage our data and model training. We also use `timm` (Wightman, 2019) for implementations of ClimaX and Stormer. All training is done on 8 NVIDIA RTX A6000 GPUs. We leverage `fp16` floating point precision in our experiments.

## F  VISUALIZATIONS

### F.1  S2S FORECASTING

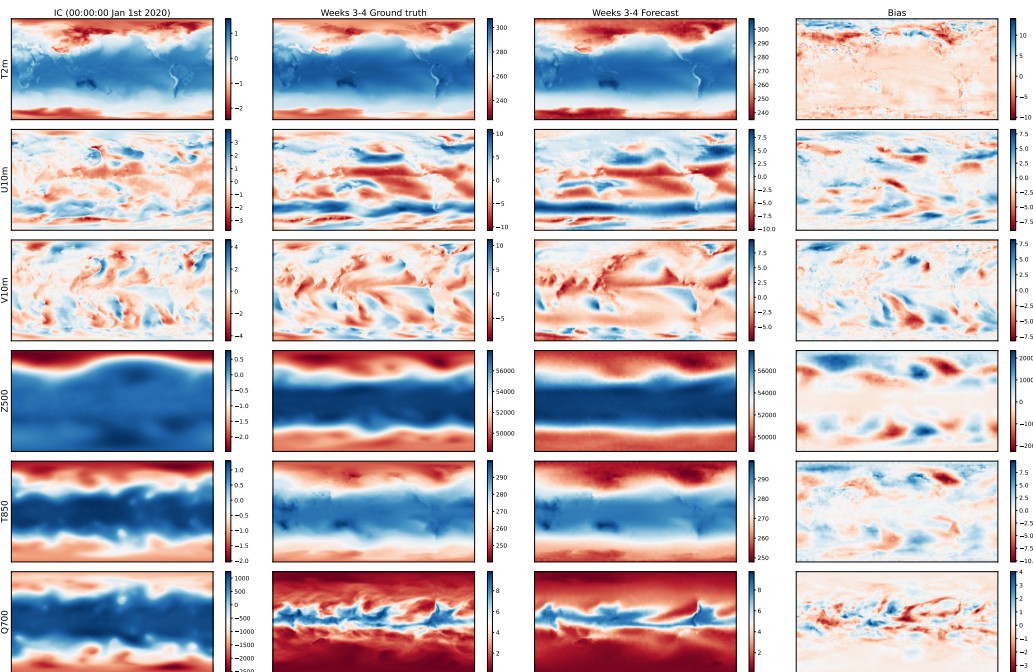

Figure 5: ClimaX forecasts for weeks 3-4 of six target variables.

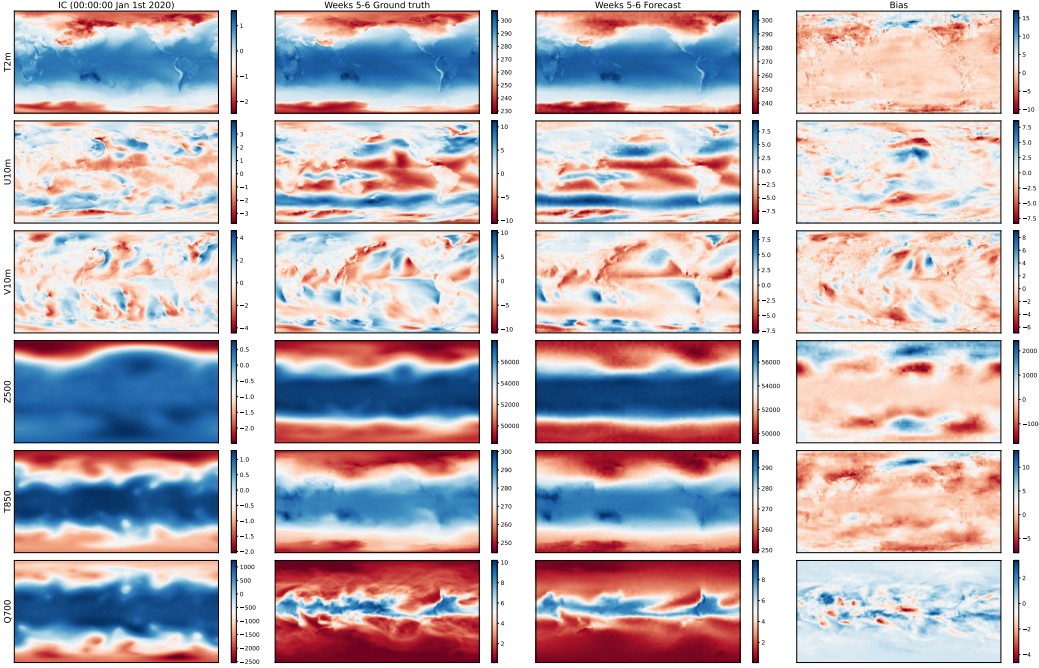

Figure 6: ClimaX forecasts for weeks 5-6 of six target variables.

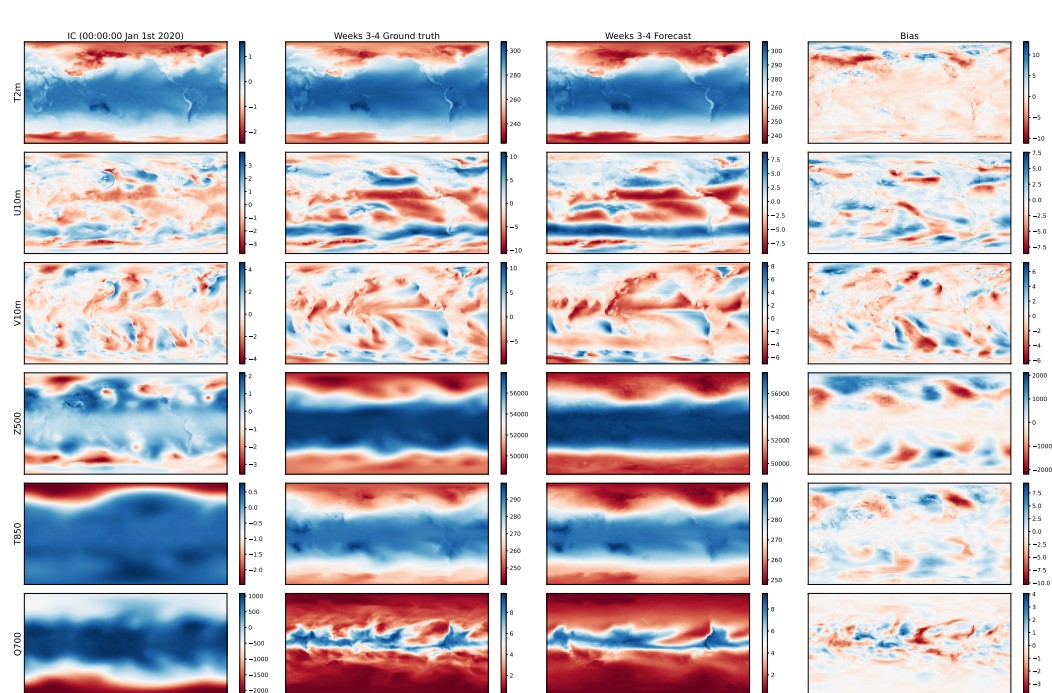

Figure 7: Stormer forecasts for weeks 3-4 of six target variables.

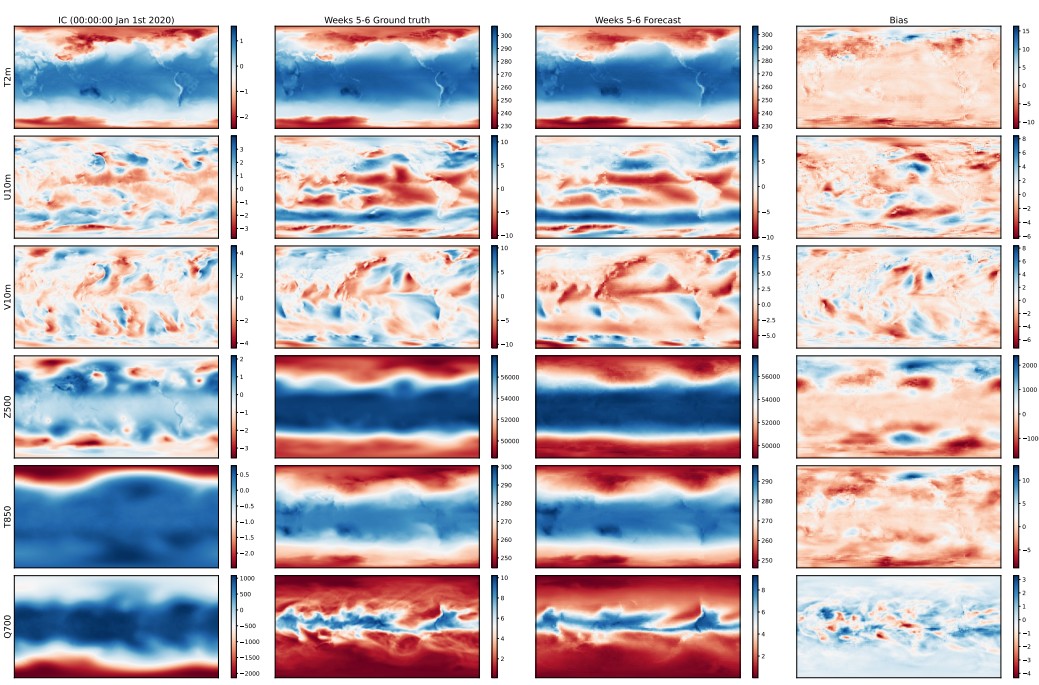

Figure 8: Stormer forecasts for weeks 5-6 of six target variables.

## F.2 CLIMATE DOWNSCALING

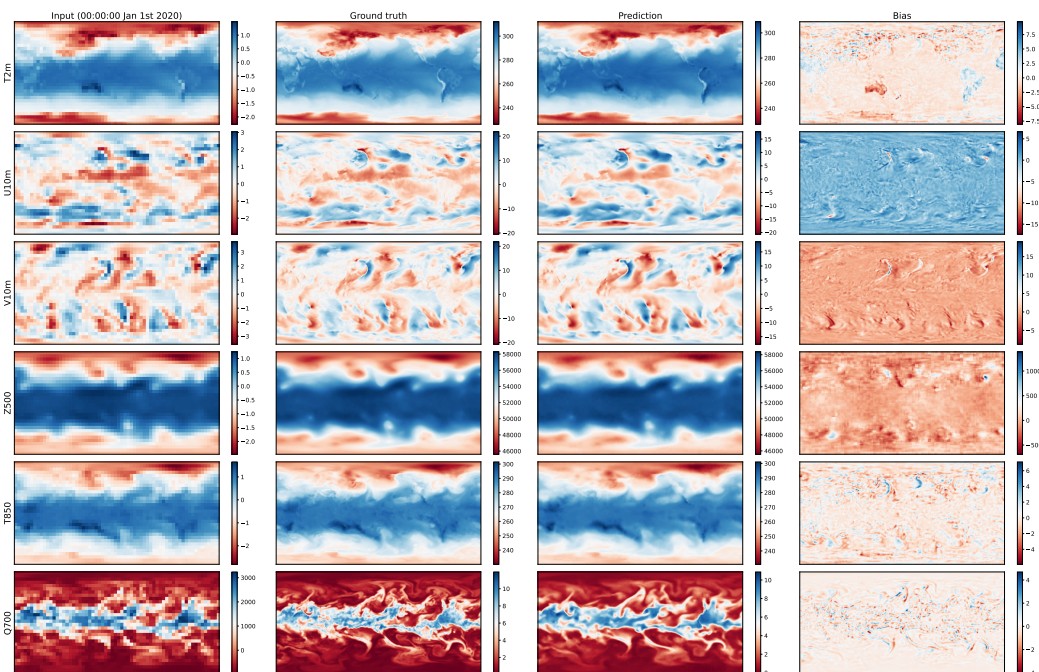

Figure 9: ClimaX downscaling predictions of six target variables.

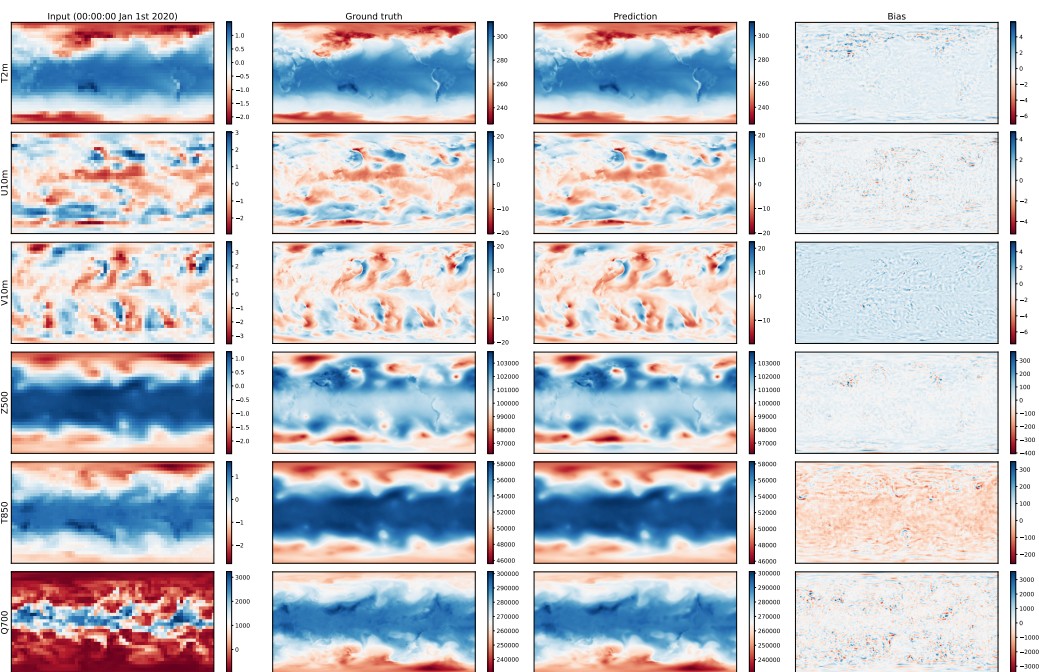

Figure 10: Stormer downscaling predictions of six target variables.

## G   ATMOSPHERIC CHEMISTRY

This section presents the atmospheric chemistry tasks that `AtmosArena` includes.

### G.1   ATMOSPHERIC CHEMISTRY DOWNSCALING

Atmospheric chemistry simulations are essential for understanding various global processes such as air pollution, biogeochemical cycles, and climate change. High-resolution models can capture fine-scale chemical interactions, providing insights into local pollution levels and their health impacts. However, these models are computationally intensive. Deep learning offers a solution by transforming coarse-resolution inputs into finer-resolution outputs (Geiss et al., 2022). Specifically, the input is a grid of dimensions $V \times H \times W$, and the output is a higher-resolution grid $V' \times H' \times W'$, where $H' > H$ and $W' > W$. This allows for precise monitoring of atmospheric pollutants and their effects on human health and the environment, enabling more informed policy decisions and scientific research.

**Dataset**   We utilize GEOS-CF, a simulated dataset from the NASA GEOS Composition Forecast (GEOS-CF) system (Knowland et al., 2022). GEOS-CF combines the NASA GEOS model with the GEOS-Chem chemical transport model to simulate the atmospheric composition (Keller et al., 2021). The dataset offers outputs on a $0.25°$ grid, which we downsample to $5.625°$ for the low-resolution input and $1.40625°$ for the high-resolution output. For our benchmark, we use the meteorological replay simulation ("das" files), covering the years 2018 to the present. We focus on downscaling the five near-surface atmospheric pollutants: NO2, SO2, CO, O3, and PM2.5, averaged over a 1-hour window ("chm_tavg_1hr" files).

### G.2   ATMOSPHERIC COMPOSITION FORECASTING

This task involves predicting the global atmospheric composition of important air pollutants such as carbon monoxide and carbon dioxide at different lead times. This task is crucial for understanding air quality, which directly impacts human health by influencing the prevalence of non-communicable diseases. This task presents a significant challenge to data-driven models due to the complexity of atmospheric dynamics and the influence of human activities on emission levels. The task formulation and input and output shapes are similar to weather forecasting.

**Dataset**   We use CAMS Analysis maintained by ECMWF for the atmospheric composition forecasting task in `AtmosArena`. As part of the Copernicus Atmosphere Monitoring Service (CAMS), this dataset integrates meteorological variables with concentrations of air pollutants such as carbon monoxide and carbon dioxide, providing a comprehensive overview of global atmospheric composition. The dataset offers 12-hourly data at a $0.4°$ ($450 \times 900$ grids) resolution from 2017 to the present. Similar to ERA5, we regrid this dataset to the common resolution of $1.40625°$ for easier training and evaluation.

## H   ADDITIONAL EXPERIMENTS

### H.1   ATMOSPHERIC CHEMISTRY EXPERIMENTS

#### H.1.1   ATMOSPHERIC CHEMISTRY DOWNSCALING

We consider the task of downscaling for five near-surface variables: NO2, SO2, CO, O3, and PM2.5. We use GEOS-CF at $5.625°$ as the low-resolution input, and GEOS-CF at $1.40625°$ as the high-resolution target, corresponding to $4\times$ upsampling. We use 2018-2020 for training, 2021 for validation, and 2020 for testing. Due to time and compute constraints, we only consider ClimaX finetuned and Unet as baselines.

Table 11 reports the MAE metric in the log space of the five target variables. ClimaX finetuned and Unet perform competitively. Given the results in climate downscaling, we believe fully finetuned Stormer will outperform Unet in this task.

Table 11: MAE of ClimaX finetuned and Unet for downscaling five target near-surface pollutants.

|                   | NO2       | SO2       | CO    | O3        | PM2.5     |
|-------------------|-----------|-----------|-------|-----------|-----------|
| ClimaX finetuned  | 0.069     | 0.049     | 0.405 | **0.0065** | **0.100** |
| Unet              | **0.064** | **0.047** |       | 0.0071    | 0.104     |

### H.1.2 ATMOSPHERIC COMPOSITION FORECASTING

We compare ClimaX with Unet on forecasting eight near-surface pollutants: Total Column Carbon Monoxide (TC CO), Total Column Nitric Oxide (TC NO), Total Column Nitrogen Dioxide (TC NO2), Total Column Sulfur Dioxide (TC SO2), Total Column Ozone (TC O3), Particulate Matter 1um (PM1), Particulate Matter 2.5um (PM2.5), and Particulate Matter 10um (PM10), with lead times from 1 to 3 days. For each baseline method, we finetune a separate model for each specific lead time and target variable. We use CAMS Analysis from 2017 to 2020 for training, 2021 for validation, and 2022 for testing.

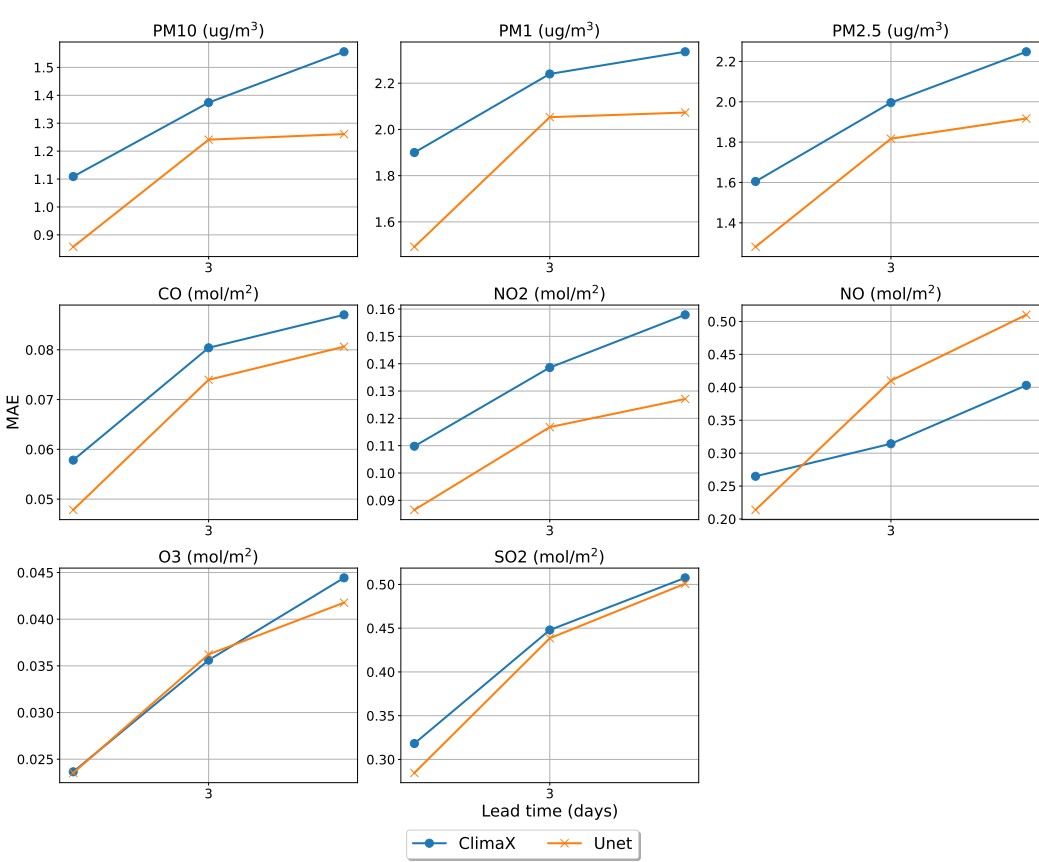

Figure 11: Air composition forecasting performance.

Figure 11 shows the performance of ClimaX and Unet on forecasting eight key pollutants from 1 day to 5 days. Unet outperforms ClimaX for almost all variables. This result shows that the temporal forecasting capabilities of pretrained models may not transfer well to new tasks in a new domain.

### H.2 ADDITIONAL METRICS FOR ATMOSPHERIC PHYSICS TASKS

**S2S forecasting** In addition to RMSE and ACC, we report Spectral Divergence as a physics-based metric, which measures the discrepancy between the frequency components of the ground truth and prediction. Table 12 shows the superior performance of ClimaX frozen across all variables and lead

times. This highlights the effectiveness of multi-source pretraining in obtaining a general-purpose backbone that can adapt to forecasting tasks with unseen time scales only via lightweight finetuning.

Table 12: S2S performance measured by Spectral Div on four target variables at two lead times.

| | | Z500 | | T850 | | T2m | | Q700 | |
|---|---|---|---|---|---|---|---|---|---|
| | | Weeks 3-4 | Weeks 5-6 | Weeks 3-4 | Weeks 5-6 | Weeks 3-4 | Weeks 5-6 | Weeks 3-4 | Weeks 5-6 |
| **Spectral Div** ($\downarrow$) | ClimaX frozen | 0 | 0 | **0.3153** | **0.2894** | **0.1671** | **0.1805** | **0.0789** | **0.0903** |
| | ClimaX finetuned | 0 | 0 | 0.3224 | 0.3180 | 0.2298 | 0.2093 | 0.0930 | 0.0937 |
| | Stormer frozen | 0 | 0 | 0.3307 | 0.4161 | 0.4705 | 0.5971 | 0.5188 | 0.7513 |
| | Stormer finetuned | 0 | 0 | 0.3275 | 0.3024 | 0.6603 | 0.6105 | 0.4337 | 0.3468 |
| | Unet | 0 | 0 | 0.3863 | 0.5110 | 0.2065 | 0.4647 | 0.0809 | 0.8157 |

**Downscaling** Table 13 shows the Anomaly Pearson Coefficient of different baselines on the climate downscaling tasks. Stormer finetuned is the best method for all four variables. However, all baselines achieve very similar performances, suggesting Anomaly Pearson Coefficient may not be the best metric for distinguishing different models in this task. A similar result was observed in ClimaX (Nguyen et al., 2023a).

Table 13: Downscaling performance measured by Anomaly Pearson Coefficient on six variables.

| | | Z500 | T850 | T2m | Q700 | U10 | V10 |
|---|---|---|---|---|---|---|---|
| **Anomaly Pearson** ($\uparrow$) | ClimaX frozen | 0.9963 | 0.9879 | 0.9833 | 0.9388 | 0.9690 | 0.9716 |
| | ClimaX finetuned | 0.9977 | 0.9907 | 0.9869 | 0.9532 | 0.9802 | 0.9813 |
| | Stormer frozen | 0.9956 | 0.9856 | 0.9821 | 0.9240 | 0.9654 | 0.9689 |
| | Stormer finetuned | **0.9993** | **0.9951** | **0.9946** | **0.9626** | **0.9886** | **0.9894** |
| | Unet | 0.9987 | 0.9931 | 0.9917 | 0.9613 | 0.9850 | 0.9861 |

**Extreme weather events detection** Table 14 shows the Specificity metrics of different methods in the extreme weather events detection tasks. ClimaX frozen is the best-performing method, showing the effectiveness of multi-source pretraining in transferring the backbone to a completely new task. However, the baselines perform very similarly for this metric, suggesting it may not be the best to evaluate methods in this task.

Table 14: Specificity Metrics of different methods for TC and AR detection.

| | ClimaX frozen | ClimaX finetuned | Stormer frozen | Stormer finetuned | CGNet |
|---|---|---|---|---|---|
| TC | 0.99 | 0.99 | 0.98 | 0.98 | 0.99 |
| AR | 0.96 | 0.96 | 0.95 | 0.95 | 0.92 |

## H.3 CLIMATE DATA INFILLING ON BERKELEY EARTH

We test the models trained to perform infilling for ERA5 in Sections 4.4 on the Berkeley Earth dataset to examine their transferability between datasets. Similarly to ERA5, we generate a fixed set of masks during testing, with the mask ratio $r \in \{0.1, 0.3, 0.5, 0.7, 0.9\}$. We test the models on infilling for data from 2020 to 2023. Figure 12 shows that all methods perform similarly for this dataset, and the performances do not get worse as we increase the mask ratio. We hypothesize that because of the distribution shift from ERA5 to Berkeley Earth, the best thing the models can do is to predict the average, leading to very similar performances among models and mask ratios.

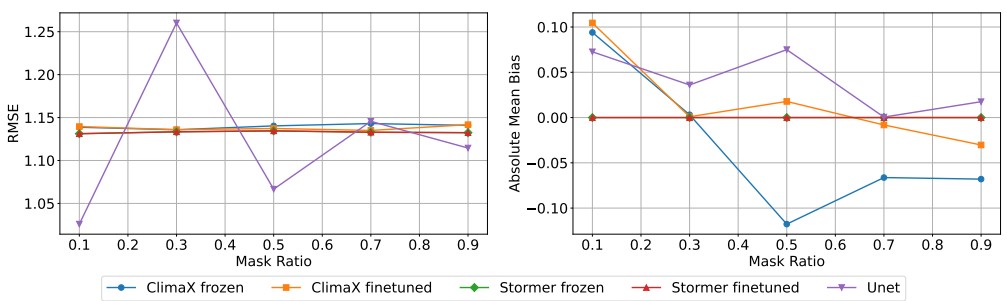

Figure 12: Performance of different models measured by RMSE and Absolute Mean Bias on infilling the Berkeley Earth temperature data with different mask ratios.

