# OpenReview forum: "AtmosArena: Benchmarking Foundation Models for Atmospheric Sciences"
_ICLR.cc/2025/Conference — Submitted to ICLR 2025_

### Official Review · Reviewer_XwUv · 2024-10-29

**Soundness:** 3
**Presentation:** 3
**Contribution:** 3
**Rating:** 6
**Confidence:** 3

**Summary:**

This article proposes a new benchmark for evaluating atmospheric and weather foundation models. It comprises the following tasks:
weather forecasting, S2S forecasting, climate data infilling, climate model emulation, climate downscaling, and extreme weather events detection with several metrics for each task. datasets, fine-tuning protocols, evaluation code, standardized metrics, and traditional and machine learning baselines.

**Strengths:**

This paper presents a standardized benchmark for climate and weather evaluation with data, metrics, code, and baselines. It evaluates all consistently, transparently, in a way that it is easy to reproduce and that will facilitate and boost the research in this area.

**Weaknesses:**

AtmosArena, does not offer a pertaining dataset, only evaluation.

It is not clear to me why we need a new benchmark. In the related work section of benchmark three should be a clear comparison with the other benchmarks and why this one is better and needed.

**Questions:**

AtmosArena has a great overlap with ClimateLearn, ClimaX, and Aurora. Almost all the tasks and datasets except for ClimateNet and Berkeley Earth are already in the other datasets. Is it that you created new annotations, or is it easier to use? Why do we need AtmosArena and not to test on the other ones directly?

---

> ### Author Response · Authors · 2024-11-22
> **Authors' rebuttal**
>
> We thank the reviewer for your constructive feedback and for recognizing our work’s consistency, transparency, and reproducibility. We address each specific concern below.
>
> > AtmosArena, does not offer a pretraining dataset, only evaluation.
>
> **We believe standardizing pretraining data is neither feasible nor desirable** - developers of atmospheric foundation models should have the flexibility to select their own datasets, resolutions, variables, and pretraining tasks, as these choices are fundamental to model development. However, all foundation models should demonstrate strong performance across diverse applications. **This mirrors established practices in NLP, computer vision, and other scientific domains like protein modeling**, where models train on varied datasets but are evaluated against common benchmarks. AtmosArena provides this standardized evaluation framework for atmospheric sciences.
>
> > Why do we need AtmosArena?
>
> Existing foundation models, including ClimaX, Aurora, and the recently published Prithvi WxC, all benchmark their models on non-overlapping sets of tasks, making comparisons difficult. AtmosArena aims to provide a single, open-source, multitask framework for benchmarking current and future foundation models, **making it both easier and fairer to compare models and assess progress in the field**. Having a standardized benchmark also helps improve expandability, since we can keep adding new tasks, datasets, and metrics to AtmosArena for evaluating future foundation models. A similar effort in NLP, **the GLUE benchmark, unified diverse language tasks and datasets proposed in other papers into a single evaluation suite, driving significant progress in the field**. AtmosArena aspires to bring the same impact to foundation models for climate and atmospheric sciences.
>
> > Comparison with ClimateLearn.
>
> **AtmosArena extends ClimateLearn significantly in many dimensions:**
> - More tasks and evaluation metrics: ClimateLearn supports 3 tasks - weather forecasting, climate downscaling, and climate projection, while AtmosArena additionally provides 5 more tasks - S2S forecasting, extreme weather events detection, climate data infilling, atmospheric chemistry downscaling, and air composition forecasting. In terms of evaluation metrics, ClimateLearn uses RMSE, ACC, and Pearson correlation, while AtmosArena additionally supports SpecDiv for forecasting, and IoU, Precision, Recall, F-1 score, and Specificity for extreme weather events detection.
> - More datasets: ClimateLearn supports 4 datasets - ERA5, CMIP6, ClimateBench, and PRISM. AtmosArena does not have PRISM, but supports Berkeley Earth for infilling, ClimateNet for extreme weather events detection, CAMS Analysis for air composition forecasting, and GEOS-CF for atmospheric chemistry downscaling. Our ERA5 dataset is also at a higher resolution (1.40625deg) compared to ClimateLearn (5.625deg).
> - Much stronger models: ClimateLearn implements three standard deep learning models, Resnet, Unet, and ViT, while AtmosArena includes ClimaX and Stormer, two strong models for atmospheric science.
>
> > Comparison with ClimaX and Aurora
>
> AtmosArena supports climate data infilling, extreme weather events detection, and atmospheric chemistry downscaling, which were not considered by either ClimaX or Aurora. In terms of datasets, we additionally support Berkeley Earth for infilling, ClimateNet for extreme weather events detection, and GEOS-CF for atmospheric chemistry downscaling. AtmosArena also provides two standard finetuning protocols to evaluate foundation models that were not studied carefully in previous works.
>
> We thank the reviewer again for the constructive feedback and continued support. We believe the reviewer’s questions and suggestions have significantly improved the paper, and **we sincerely hope that the reviewer can take our discussion into account** in the final assessment of the paper.

---

> > ### Author Response · Authors · 2024-11-24
> > **Reminder of our rebuttals**
> >
> > Dear Reviewer XwUv,
> >
> > We appreciate that you are likely to review multiple other papers; however, as we approach the end of the discussion period (two days left), we would greatly appreciate your feedback on our rebuttals. We believe our rebuttals have addressed the reviewer's two main points: explain the need for AtmosArena as a unified benchmark for atmospheric foundation models, and elaborate on the contributions of AtmosArena compared to previous works like ClimateLearn, Aurora, and ClimaX. We are happy to address any remaining concerns or questions to improve the paper further.
> >
> > Kind regards,
> > The authors of paper 12939

---

> > > ### Comment · Reviewer_XwUv · 2024-11-26
> > > **Response**
> > >
> > > Thanks to the reviewers for their clarifications. I also appreciate the effort made to address the other reviewers' concerns. Selecting, standardizing, extending, and improving datasets is helpful for the community and requires a lot of work. I appreciate that you have built upon existing benchmarks and improved them. Accordingly, I stand with my score of marginally above the acceptance threshold.

---

> > > > ### Author Response · Authors · 2024-11-26
> > > > **Thank you**
> > > >
> > > > We thank the reviewer for the constructive feedback and for recognizing the contribution of our work to the community.

---

### Official Review · Reviewer_9H2z · 2024-10-31

**Soundness:** 2
**Presentation:** 3
**Contribution:** 2
**Rating:** 6
**Confidence:** 4

**Summary:**

The authors propose a benchmark suite in which they collect several benchmark datasets into one framework which therefore offers a broader set of tasks to evaluate foundation models in weather and climate. Using their AtmosArena setup, the authors evaluate two prominent foundation models, ClimaX and Stormer on the set of atmospheric phsysical tasks and highlight performance
differences that underline the usefulness of the benchmark.

**Strengths:**

The set of tasks under this new benchmark suite is larger than any of the individual existing frameworks and therefore offers the chance for a more in depth comparison of foundation models across tasks. The paper is clearly written and structured such that it is easy to follow.

**Weaknesses:**

In my opinion a shortcoming of the proposed framework is that it is more a collection of existing benchmark frameworks like Weatherbench, ClimateBench, and ClimateNet and misses some opportunities for improvement. This statement to me is not about "novelty", but rather remaining shortcomings or inconsistencies under this new framework.

- While you state that in the future you would like to have a public
  leaderboard, I think this is already something to be expected from such a
  framework. Something like the google [Weatherbench leaderboard](https://sites.research.google/weatherbench/) is a decent
  refrenece to get a first idea of perfomrance across models
- For the task of climate down scaling you employ the RMSE, Bias and Pearson
  correlation as available metrics, however, especially here, metrics like Power
  Spectral Density (PSD), and PSD plots are important and offer more insights than a single error metric value
-  in fact, for all climate related prediction tasks (Forecasting, Super-resolution, inpainting mentioned in your paper) spectral metrics are
  commonly employed in the atmospheric science literature and should be a part of an atmospheric benchmark
- as another point, I think not including any probabilistic metrics across the
  collected tasks is unfortunate because simple point estimates just come short
  of the complexities given these tasks. A notion of prediction uncertainty and
  an assessment of the predictive uncertainty with metrics like proper scoring
  rules seems essential when aiming to do an holistic analysis. Probabilistic metrics were for example included in Weatherbench2 but seem to be missing from your framework.

Given that from my understanding you have collected existing frameworks under a new benchmark suite, I would have expected some additional improvements about the shortcomings of those, especially surrounding additional evaluation metrics as this is such an important part of benchmarking. While the employed metrics are very common in machine learning, I think additional metrics like PSD that exist in the atmospheric science literature are essential for a good benchmark framework.

**Questions:**

Paper Questions:
- in section 4.5, Table 4, the numbers reported for ClimaX, ClimaX frozen and
  ClimateBench-NN are exactly the same as the values reported in the ClimaX
  publication Table 2, which I do find a bit surprising that there is no
  statistical difference given the non-deterministic nature of Deep Learning
  model training and a rather small data size
- is there a reference for the Spectral Divergence metric you use, or is this a metric you propose in this work?
- Line 18, you say the "first multi-task benchmark", but in table 1 you list
  existing works that consider "multiple atmospheric tasks" so I don't think
  your claim of "first" holds, but maybe you can clarify?

General Comment:
- I believe a work like this is highly dependent on the quality of the code,
  this is difficult to assess in this review process which is very unfortunate

---

> ### Author Response · Authors · 2024-11-22
> **Authors' Rebuttal**
>
> We thank the reviewer for your constructive feedback and for appreciating the large set of tasks AtmosArena offers and the clear writing of the paper. We address each specific concern below.
>
> > While you state that in the future you would like to have a public leaderboard, I think this is already something to be expected from such a framework.
>
> We thank the reviewer for this suggestion. **We created a leaderboard at https://atmosarena.github.io/leaderboard/ for the tasks we consider in this work**. The leaderboard can be expanded in the future to include more models, tasks, and evaluation metrics. We have also added a link to the leaderboard to the paper.
>
> > Metrics like Power Spectral Density (PSD), and PSD plots are important and offer more insights than a single error metric value
>
> We thank the reviewer for this great suggestion. **We have added the PSD plots for the downscaling task in the updated submission**. To create these plots, we computed the 2D Power Spectral Density using the Fast Fourier Transform (FFT) for each spatial field, then performed radial averaging to obtain 1D PSD curves that show how power varies with spatial frequency. For each variable (T2M, Z500, T850), we plotted the PSD curves of the ground truth and predictions from three models (ClimaX, Stormer, UNet) on a log-log scale.
>
> The PSD plots (Figure 2 in the paper) show that all three models show excellent agreement with the ground truth across low to medium spatial frequencies for all variables, indicating they accurately capture large-scale spatial patterns. However, there are notable differences at high spatial frequencies (>0.2): UNet tends to underestimate the power at these frequencies, suggesting it may smooth out fine-scale details, while ClimaX and Stormer better preserve these high-frequency components. The results suggest that two pretrained models, ClimaX and Stormer, have an advantage in preserving fine-scale spatial details compared to UNet.
>
> > I think not including any probabilistic metrics across the collected tasks is unfortunate because simple point estimates just come short of the complexities given these tasks
>
> We provided the probabilistic metrics, including CRPS and Spread-skill ratio in the source code. **We did not include any probabilistic metrics in the paper only because all models we consider: ClimaX, Stormer, and UNet, are deterministic models**. Even the two newer foundation models Aurora and Prithvi WxC are deterministic. We believe this is not a limitation of our evaluation framework but rather a limitation of existing models. We can always benchmark future models on uncertainty metrics if they provide probabilistic predictions.
>
> > I would have expected some additional improvements about the shortcomings of those, especially surrounding additional evaluation metrics
>
> In addition to the PSD plots the reviewer suggested, we also provide Spectral Divergence (SpecDiv), a physics-based metric not considered by previous works.
>
> > In section 4.5, Table 4, the numbers reported for ClimaX, ClimaX frozen and ClimateBench-NN are exactly the same as the values reported in the ClimaX publication Table 2
>
> Finetuning ClimaX is expensive because we have to finetune one model for each target variable in ClimateBench, so we used the model checkpoints provided by the ClimaX authors as is. Since the model itself is deterministic and we used the same train and test splits as ClimaX, we were able to reproduce the numbers.
>
> > Is there a reference for the Spectral Divergence metric you use, or is this a metric you propose in this work?
>
> To the best of our knowledge, Spectral Divergence was first proposed for image analysis [1] and ChaosBench [2] was the first paper that used it for weather and climate.
>
> [1] Chang, Chein-I. "Spectral information divergence for hyperspectral image analysis." IEEE 1999 International Geoscience and Remote Sensing Symposium. IGARSS'99 (Cat. No. 99CH36293). Vol. 1. IEEE, 1999.
>
> [2] Nathaniel, Juan, et al. "Chaosbench: A multi-channel, physics-based benchmark for subseasonal-to-seasonal climate prediction." arXiv preprint arXiv:2402.00712 (2024).
>
> > I believe a work like this is highly dependent on the quality of the code
>
> **We have submitted the source code as supplementary material**. The code has detailed instructions on reproducing the experiments in the paper. We also developed individual components such as models, data, and metrics in a way that can be used independently.
>
> We thank the reviewer again for the constructive feedback and continued support. We spent significant efforts in providing additional metrics, published source code, and the public leaderboard, which we believe have significantly improved the paper. **We sincerely hope that the reviewer can take our effort during the rebuttal into account** in the final assessment of the paper.

---

> > ### Author Response · Authors · 2024-11-24
> > **Reminder of our rebuttals**
> >
> > Dear Reviewer 9H2z,
> >
> > We appreciate that you are likely to review multiple other papers; however, as we approach the end of the discussion period (two days left), we would greatly appreciate your feedback on our rebuttals. We believe our rebuttals have addressed the reviewer's three main points: 1) create a leaderboard for the tasks supported by AtmosArena, 2) add PSD plots for the downscaling task, and 3) submit the source code to reproduce the results in AtmosArena. We spent significant time and effort on the additional results and analysis and we are happy to address any remaining concerns or questions to improve the paper further.
> >
> > Kind regards,
> > The authors of paper 12939

---

> ### Comment · Reviewer_9H2z · 2024-11-25
> **Rebuttal**
>
> I thank the authors for the additional effort they have put in, especially regarding the leaderboard. However, I have some remaining remarks:
>
> Regarding the PSD metrics, I apologize for being unclear. My point was not specific to climate downscaling as PSD metrics are relevant for all sorts of other spatio-temporal prediction tasks some of which are included in your framework, such as precipitation prediction in climate emulation. My goal is to underline that the evaluation needs to be more holistic than what the Machine Learning lense currently does, predominantly using distance RMSE metrics. For example your updated manuscript regarding Climate Downscaling reads in line 406: " Stormer is the best model in this task with the lowest RMSE and Absolute Mean Bias for most variables, followed by the Unet baseline". But is that the case if you consider the PSD plots? I am not saying that your task is to find the best model, but I would argue that evaluation needs to be a lot more nuanced.
>
> Regarding probabilistic metrics: I understand your point about the models being deterministic. While you argue rightfully that this is a shortcoming of the models, I would also argue that a comprehensive benchmarking framework should "expose" such shortcomings, because a benchmark should not just test what models can already do but set a higher bar. For example Table 1 could include the probabilistic metrics like CRPS with a note that none of the models are able to do probabilistic forecasts. But at the moment the word "uncertainty" appears once in your manuscript, and with no probabilistic metric mentioned or discussion about uncertainty in forecasting or benchmarking, I would argue that this "weakness" remains.
>
> Regarding the quality of the code: Thank you for including the source code in the zip. I did not expect a discussion about code quality but at the minimum I would argue that for a community benchmark framework, it should include docstrings for all functions and classes which seems to be the case for some but not others. I attempted to run the conda installation and code but got:
>
> ```
> Pip subprocess error:
> ERROR: Could not find a version that satisfies the requirement enrich==0.1.dev82 (from versions: 0.1.dev1, 1.0, 1.1, 1.2, 1.2.1, 1.2.2, 1.2.3, 1.2.4, 1.2.5, 1.2.6, 1.2.7)
> ERROR: No matching distribution found for enrich==0.1.dev82
>
> failed
>
> CondaEnvException: Pip failed
> ```
> on Ubuntu 20.04.
>
> I would like to update my score to 5 to reflect the additional work that the authors have put in, however, would not change my score to "acceptance".

---

> > ### Author Response · Authors · 2024-12-02
> >
> > We thank the reviewer for your feedback on our rebuttals. We address the remaining concerns below.
> >
> > > PSD metrics are relevant for all sorts of other spatio-temporal prediction tasks
> >
> > We agree. Indeed, **the PSD metrics in AtmosArena are applicable across all tasks in our benchmark suite. Our presentation of PSD plots for climate downscaling serves to demonstrate that our benchmark framework fully supports these metrics.** We will include an appendix section in the revised manuscript showing PSD metrics for other tasks.
> >
> > > For example your updated manuscript regarding Climate Downscaling reads in line 406: " Stormer is the best model in this task with the lowest RMSE and Absolute Mean Bias for most variables, followed by the Unet baseline". But is that the case if you consider the PSD plots?
> >
> > Figure 2 in our paper shows that there is no clear winner in terms of PSD metrics, and the ranking of the baselines varies across different data samples and. variables. Given that Stormer outperforms the other two baselines significantly in RMSE, we can still conclude that Stormer is the best method for this task.
> >
> > > I would also argue that a comprehensive benchmarking framework should "expose" such shortcomings
> >
> > We will add the CRPS to Table 1 and discuss the lack of uncertainty estimation of existing methods in the updated manuscript. However, **this does not present a weakness of our benchmark, but rather limitations of existing methods.**
> >
> > > I would argue that for a community benchmark framework, it should include docstrings for all functions and classes which seems to be the case for some but not others. I attempted to run the conda installation and code but got
> >
> > **We respectfully disagree with judging the submission based on code documentation at this stage.** The paper should be evaluated on its primary contributions: establishing a benchmark framework for atmospheric foundation models, defining comprehensive tasks and metrics, and demonstrating the framework's capabilities through extensive experiments. While we acknowledge the importance of complete code documentation, refinement of the codebase - including improved docstrings - will naturally follow publication, as is standard practice in the field. The scientific merits of the paper itself should be the focus of the current evaluation.

---

### Official Review · Reviewer_xs49 · 2024-11-02

**Soundness:** 3
**Presentation:** 3
**Contribution:** 3
**Rating:** 6
**Confidence:** 4

**Summary:**

The authors present a comprehensive collection of benchmarks, with appropriate baseline models, for tasks related to atmospheric science and prediction.

**Strengths:**

While each component of the dataset is not novel, the combination and the testing of multi-model foundational models make this a useful contribution. The baselines are well chosen and the data appears to be well structured (though I have not tested this). The paper is well structured and well written. In particular, each of the sub-tasks defined in the 'arena' are appropriate and complementary, and the baselines are appropriate and test interesting aspects of their generalizability.

**Weaknesses:**

It isn't entirely clear that any of the presented models are foundation models; it feels like the 'atmosphere' is a bit too specific of a system to qualify as foundational. To some extent this is something the cited models need to demonstrate, but nonetheless, to demonstrate its utility, this paper should describe what they mean by a foundation model for the atmosphere, and why the tasks they present (within a fairly narrow range of spatial and temporal scales) would test such foundational knowledge.  Relatedly, the paper feels a bit like a collection of benchmarks and could do a better job of explaining why the various tasks are complementary, e.g. because they all rely on some underlying understanding of the covariance of the atmosphere over different spatial and temporal scales (be explicit).

**Questions:**

Please explicitly describe how the presented tasks complement each other and aim to test the necessary variables, and spatial and temporal scales to qualify a model as 'foundational'.

---

> ### Author Response · Authors · 2024-11-22
> **Authors' Rebuttal**
>
> We thank the reviewer for your constructive feedback and for appreciating the contributions, well-chosen data, tasks, and baselines of AtmosArena, and the clear writing of our paper. We address each specific concern below.
>
> > It isn't entirely clear that any of the presented models are foundation models; it feels like the 'atmosphere' is a bit too specific of a system to qualify as foundational.
>
> **What we mean by “foundation models for the atmosphere” is models that can solve various tasks in atmospheric sciences**, including forecasting, downscaling, extreme weather events detection, etc. We adopt the term “foundation” similarly to how people have been referring to GPT-x or Llama as language foundation models or Stable Diffusion as an image foundation model. **Foundation models are also ubiquitous in other scientific fields** such as EVO for genome, Prithvi for Earth science and remote sensing, AstroCLIP for astrophysics, etc. The fact that both ClimaX and Stormer outperform the non-pretrained baseline UNet in many tasks shows their capability of generalizing to different downstream tasks.
>
> > The paper could do a better job of explaining why the various tasks are complementary, e.g. because they all rely on some underlying understanding of the covariance of the atmosphere over different spatial and temporal scales (be explicit).
>
> **The tasks we chose are complementary from both atmospheric and machine learning perspectives:**
> - From a domain perspective, atmospheric tasks are broadly categorized into atmospheric physics or atmospheric chemistry, and AtmosArena supports both categories.
> - From a machine learning perspective, tasks in AtmosArena are mapped to well-defined problems in machine learning: forecasting, segmentation, super-resolution, inpainting, and counterfactual prediction. As the reviewer mentioned above, these tasks together test a model’s understanding of the atmosphere over different spatial and temporal scales.
>
> We thank the reviewer again for the constructive feedback and continued support. We believe the reviewer’s questions and suggestions have significantly improved the paper, and **we sincerely hope that the reviewer can take our discussion into account** in the final assessment of the paper.

---

> > ### Author Response · Authors · 2024-11-24
> > **Reminder of our rebuttals**
> >
> > Dear Reviewer xs49,
> >
> > We appreciate that you are likely to review multiple other papers; however, as we approach the end of the discussion period (two days left), we would greatly appreciate your feedback on our rebuttals. We believe our rebuttals have addressed the reviewer's two main points: clarify what foundation models for atmospheric sciences mean and why the tasks we chose in the paper are complementary from both machine learning and atmospheric science perspectives. We are happy to address any remaining concerns or questions to improve the paper further.
> >
> > Kind regards,
> > The authors of paper 12939

---

> > > ### Comment · Reviewer_xs49 · 2024-11-25
> > > **Response**
> > >
> > > Thank you for taking the time to respond to my queries. I agree that the ML tasks are fairly comprehensive but disagree that any of the tasks can be categorized as atmospheric chemistry. The tasks labelled as such are just downscaling or forecasting other atmospheric tracers, which are transported physically. I appreciate that the term 'foundation model' has been taken up in other domains but still disagree that the tasks represented here (or performed by e.g. ClimaX) represent the same level of generalization as e.g. LLMs. I feel the benchmark is a useful iteration on other benchmarks, but nothing more and stand by my score.

---

> > > > ### Author Response · Authors · 2024-11-26
> > > > **Thank you**
> > > >
> > > > We thank the reviewer for the constructive feedback and for recognizing the useful contribution of our work. We will keep expanding the benchmark with new tasks, datasets, and models to improve its diversity and applicability.

---

### Official Review · Reviewer_eLRx · 2024-11-04

**Soundness:** 3
**Presentation:** 3
**Contribution:** 3
**Rating:** 5
**Confidence:** 3

**Summary:**

The paper introduces AtmosArena, a benchmark designed for evaluating foundation models in atmospheric sciences, focusing on multi-task spatiotemporal learning. The paper mainly evaluates two prominent models, ClimaX and Stormer, comparing their performance with traditional baselines across various tasks. Stormer performs well in short-term forecasting, while ClimaX excels in tasks with longer temporal horizons, highlighting the benefits of multi-source pretraining. Overall, the authors demonstrate that pre-trained models generally outperform task-specific baselines, and AtmosArena serves as a comprehensive tool for advancing atmospheric modeling.

**Strengths:**

1. **New Perspective:** The article evaluates atmosphere modeling from a multitasking perspective and verifies the effectiveness of two pre-trained models: ClimaX (multi-source pre-trained models) and Stormer (single-source pre-trained models).
2. **Open Source:** By providing a standardized and open-source framework, AtmosArena sets a new standard for reproducibility and transparency in multi task atmospheric learning.
3. **Include Finetuneing:** The paper explores the finetuning protocols for foundation models, comparing frozen versus fully finetuned backbones.
4. **Diverse Tasks:** AtmosArena benchmark includes both atmospheric physics and atmospheric chemistry tasks, and the benchmarks utilize well-regarded datasets like ERA5, ClimateBench, and ClimateNet.

**Weaknesses:**

1. Regarding the completeness of the benchmark, I believe the authors should add relevant metrics, such as **inference FLOPs and model parameters**.

2. The authors need to provide more **evidence (e.g., frameworks and code)** to demonstrate the benchmark’s ease of use and reproducibility.

3. Although the authors conducted extensive experiments to show that no single model excels across all tasks, I believe they should **further analyze** the experimental results rather than simply testing performance across different tasks.

**Questions:**

See the weaknesses.

---

> ### Author Response · Authors · 2024-11-22
> **Authors' Rebuttal**
>
> We thank the reviewer for your constructive feedback and for recognizing the novelty, reproducibility, and diversity of tasks AtmosArena offers. We address each specific concern below.
>
> > The authors should add relevant metrics, such as inference FLOPs and model parameters
>
> **We have added the inference FLOPs and model parameters of the three baselines (ClimaX, Stormer, and UNet) to the updated submission**. The table below summarizes the numbers.
>
> | | ClimaX | Stormer | UNet |
> |---|---|---|---|
> | FLOPs | 986.098B | 7377.751B | 969.404B |
> | Parameter count | 110.842M | 468.752M | 577.745M |
>
> > The authors need to provide more evidence (e.g., frameworks and code) to demonstrate the benchmark’s ease of use and reproducibility.
>
> **We have submitted the code as supplementary material**. The code provides concrete instructions on reproducing the paper's results and using individual components like data and metrics.
>
> > Although the authors conducted extensive experiments to show that no single model excels across all tasks, I believe they should further analyze the experimental results rather than simply testing performance across different tasks.
>
> We believe one important aspect of a multitask benchmark like AtmosArena is to compare the performance of different methods across a diverse set of tasks. Through comparisons in diverse tasks and settings, we have also observed interesting and consistent patterns that are useful for future development of foundation models:
> - ClimaX tends to perform better in forecasting at longer lead times (S2S scale) than Stormer which is explained by the wider range of lead times used for pretraining ClimaX.
> - Multi-source pretrained models like ClimaX tend to perform better at downstream tasks that are significantly different from pretraining, including climate model emulation and extreme weather events detection.
> - Freezing the transformer backbone tends to work better in tasks with little data, which demonstrates the transferability of pretrained backbones to downstream tasks.
>
> If you have any specific analysis in mind, please let us know. We are glad to run them during the rebuttal phase.
>
> We thank the reviewer again for the constructive feedback and continued support. We believe the additional metrics and the published source code have significantly improved the paper, and **we sincerely hope that the reviewer can take our effort during the rebuttal into account** in the final assessment of the paper.

---

> > ### Author Response · Authors · 2024-11-24
> > **Reminder of our rebuttals**
> >
> > Dear Reviewer eLRx,
> >
> > We appreciate that you are likely to review multiple other papers; however, as we approach the end of the discussion period (two days left), we would greatly appreciate your feedback on our rebuttals. We believe our rebuttals have addressed the reviewer's two main points: support additional metrics like FLOPs and model parameters and provide the source code. We are happy to address any remaining concerns or questions to improve the paper further.
> >
> > Kind regards,
> > The authors of paper 12939

---

> > > ### Author Response · Authors · 2024-11-26
> > > **Reminder of our rebuttals**
> > >
> > > Dear Reviewer eLRx,
> > >
> > > We appreciate that you are likely to review multiple other papers; however, as we approach the end of the discussion period, we would greatly appreciate your feedback on our rebuttals. We believe our rebuttals have addressed the reviewer's two main points: support additional metrics like FLOPs and model parameters and provide the source code. We are happy to address any remaining concerns or questions to improve the paper further.
> > >
> > > Kind regards, The authors of paper 12939

---

> > > > ### Author Response · Authors · 2024-11-30
> > > > **Reminder of our rebuttals**
> > > >
> > > > Dear Reviewer eLRx,
> > > >
> > > > It's been more than a week since we posted our rebuttal. We appreciate that you are likely to review multiple other papers; however, as we approach the end of the discussion period, we would greatly appreciate your feedback on our rebuttals. Our rebuttals have addressed the reviewer's two main points: support additional metrics like FLOPs and model parameters and provide the source code. We are happy to address any remaining concerns or questions to improve the paper further.
> > > >
> > > > Kind regards,
> > > > The authors of paper 12939

---

> > > > > ### Author Response · Authors · 2024-12-01
> > > > > **Reminder of our rebuttals**
> > > > >
> > > > > Dear Reviewer eLRx,
> > > > >
> > > > > It's been more than a week since we posted our rebuttal. We appreciate that you are likely to review multiple other papers; however, as we approach the end of the discussion period, we would greatly appreciate your feedback on our rebuttals. Our rebuttals have addressed the reviewer's two main points: support additional metrics like FLOPs and model parameters and provide the source code. We are happy to address any remaining concerns or questions to improve the paper further.
> > > > >
> > > > > Kind regards,
> > > > > The authors of paper 12939

---

> > > > > > ### Author Response · Authors · 2024-12-02
> > > > > > **Reminder of our rebuttals**
> > > > > >
> > > > > > Dear Reviewer eLRx,
> > > > > >
> > > > > > It's been 10 days since we posted our rebuttal. **As today is the last day the reviewer can respond to us, we sincerely hope the reviewer will provide feedback on our rebuttals soon because it is critically important to our work.** Our rebuttals have addressed the reviewer's two main points: support additional metrics like FLOPs and model parameters and provide the source code. We are happy to address any remaining concerns or questions to improve the paper further.
> > > > > >
> > > > > > Kind regards,
> > > > > > The authors of paper 12939

---

### Comment · Area_Chair_dXHZ · 2024-11-24

Dear Reviewers,

This is a gentle reminder that the authors have submitted their rebuttal, and the discussion period will conclude on November 26th AoE. To ensure a constructive and meaningful discussion, we kindly ask that you review the rebuttal as soon as possible and verify if your questions and comments have been adequately addressed.

We greatly appreciate your time, effort, and thoughtful contributions to this process.

Best regards,
AC

---

### Meta-Review · Area_Chair_dXHZ · 2024-12-16

**Metareview:**

The authors propose a new benchmark to evaluate atmosphere foundation models. The benchmark evaluates models on different tasks on 4 already existing data sources and measure a wide variety of metrics.

Reviewers found this to be valuable for the research community in order to measure progress in the field of atmospheric sciences.

Two important weaknesses for the reviewers were:
* The lack of code (eLRx, 9H2z)
* The significance of the contributions with respect to previous benchmarks. For example, eLRx requests for further analysis of the results
 xs49 asks for the contribution of this work besides being a “collection of benchmarks”, the lack of diversity of the tasks (9H2z), and XwUv
requests a better explanation on the differentiating factor of the proposed benchmark.

The authors provided the code during the rebuttal, but reviewer 9H2z was unable to install the dependencies, raising some concerns about
the state of the code.

With respect to the contribution of this work, in response to 9H2z, the authors added two additional metrics:

PSD plots and Spectral Divergence (SpecDiv). And further explanation of the differences with respect to previous evaluations in response
to XwUv.

**Overall** this work is clearly in the borderline, even considering a score update by eLR, who did not engage in discussion, from 5 to 6 and
given remaining concerns by 9H2z during the reviewer / AC discussion, I cannot recommend the acceptance of this work. However, I encourage the authors to polish the paper, improve the codebase, and make the arena more usable.

**Additional Comments On Reviewer Discussion:**

During the reviewer / AC discussion 9H2z expressed concerns about not able to even install dependencies and the fact that this work is too oriented towards error metrics.

---

### Decision · Program_Chairs · 2025-01-22

Reject